# In vivo guiding nitrogen-doped carbon nanozyme for tumor catalytic therapy

Kelong Fan[1], Juqun Xi[2], Lei Fan[3], Peixia Wang[1,4], Chunhua Zhu[2], Yan Tang[2], Xiangdong Xu[3], Minmin Liang[1], Bing Jiang[1,4], Xiyun Yan[1,4] & Lizeng Gao[2]

Nanomaterials with intrinsic enzyme-like activities (nanozymes), have been widely used as artificial enzymes in biomedicine. However, how to control their in vivo performance in a target cell is still challenging. Here we report a strategy to coordinate nanozymes to target tumor cells and selectively perform their activity to destruct tumors. We develop a nanozyme using nitrogen-doped porous carbon nanospheres which possess four enzyme-like activities (oxidase, peroxidase, catalase and superoxide dismutase) responsible for reactive oxygen species regulation. We then introduce ferritin to guide nitrogen-doped porous carbon nanospheres into lysosomes and boost reactive oxygen species generation in a tumor-specific manner, resulting in significant tumor regression in human tumor xenograft mice models. Together, our study provides evidence that nitrogen-doped porous carbon nanospheres are powerful nanozymes capable of regulating intracellular reactive oxygen species, and ferritinylation is a promising strategy to render nanozymes to target tumor cells for in vivo tumor catalytic therapy.

---

[1] Key Laboratory of Protein and Peptide Pharmaceuticals, CAS-University of Tokyo Joint Laboratory of Structural Virology and Immunology, Institute of Biophysics, Chinese Academy of Sciences, Beijing 100101, China. [2] Department of Pharmacology, School of Medicine, Institute of Translational Medicine, Yangzhou University, Yangzhou 225001, China. [3] School of Chemistry and Chemical Engineering, Yangzhou University, Yangzhou 225002, China. [4] University of Chinese Academy of Sciences, 19A Yuquan Road, Beijing 100049, China. These authors contributed equally: Kelong Fan, Juqun Xi. Correspondence and requests for materials should be addressed to X.Y. (email: yanxy@ibp.ac.cn) or to L.G. (email: lzgao@yzu.edu.cn)

Since the first evidence of ferromagnetic nanoparticles as peroxidase mimetics was reported in 2007[1], various nanomaterials have been identified that possess intrinsic enzyme-like activities[2–6]. These materials catalyze enzymatic reactions and follow similar enzymatic kinetics and mechanisms of natural enzymes under physiological conditions. Thus, the term "Nanozyme" was coined to describe the properties of this emerging generation of enzyme mimetics or artificial enzymes[7]. Nanozymes exhibit high enzymatic activity which is tunable via size control, doping and surface modification. In addition, they have multiple functions, high stability and are easy to be scaled up with low cost[8]. These advantages make them superior to natural enzymes or traditional enzyme mimetics in practical applications.

In the last decade, nanozymes have been widely used in biomedical applications including immunoassays, biosensors, antibacterial and antibiofilm agents[9,10]. Currently, considerable efforts are being made to explore the feasibility of applying nanozymes to in vivo clinical diagnosis and therapy[8,11] A critical question for in vivo applications is that how to control a nanozyme to selectively perform the desired activity, as the off-target activity leads to counter productive to the main desired activity. For instance, iron oxide nanoparticles (IONPs) possess peroxidase-like activity (increase reactive oxygen species (ROS)) under acidic pH and catalase-like activity (scavenge ROS) in neutral condition. When applying for therapy with ROS on demand, the former activity is favored and the latter one should be inactivated. This selectivity in activity requires IONPs to locate in the specific cell compartment with the appropriate pH conditions. However, no effective way has been reported to control nanozyme performance under in vivo condition. Therefore, it is necessary to develop a strategy to coordinate the nanozyme for optimal functioning upon entering of the nanozyme into its target cell.

High activity is one of the prerequisites for nanozymes to perform the desired in vivo performance. To achieve nanozymes

with high activity, an efficient way is to re-examine the enzyme-like property of nanocatalysts which have been well exploited with highly catalytic properties in chemical reactions, as enzyme-like activity and chemical catalysis may share the similar mechanism. However, there are few studies focusing on the correlation between chemical catalysis and enzymatic property in these nanocatalysts. In particular, the nitrogen-doped carbon nanomaterials (N-CNMs) are promising metal-free electrocatalysts for using in energy storage, conversion devices and water splitting due to their excellent catalytic performance[12–16], but the enzymatic property of N-CNMs has been barely investigated. In these nanomaterials, nitrogen is inserted into the lattice of the graphite structure of carbon nanomaterials. In general, these materials can reduce oxygen into various oxides and water via oxygen reduction reaction (ORR) and inversely generate oxygen via oxygen evolution reaction (OER) under extreme reaction conditions. For instance, at acidic (pH 1–2) or basic condition (pH 12), they exhibit high catalytic activity and durability when reacting with oxygen ($O_2$) via a four-electron mechanism or a 2 + 2 electron pathway accompanied by intermediate radical species[17,18]. The ratio and forms of nitrogen play a crucial role in these electron pathways. Studies on the mechanisms of N-CNMs in ORR and OER indicate that pyridinic N and quaternary N are potential active sites, although in-depth investigations are still required to clarify their roles in ORR and OER[19,20]. More specifically, the porosity of these materials improves their activity, as the porous interface provides numerous active sites and favors mass transport in the catalytic process. Thus, we speculated that N-CNMs possess high enzyme-like activities acting on active oxygen-related reactions for ROS regulation.

Here we present a successful paradigm of guiding a nanozyme to target tumor cell for in vivo tumor catalytic therapy using N-doped porous carbon nanospheres (N-PCNSs) as well-established model nanozymes. We identify that N-PCNSs are a

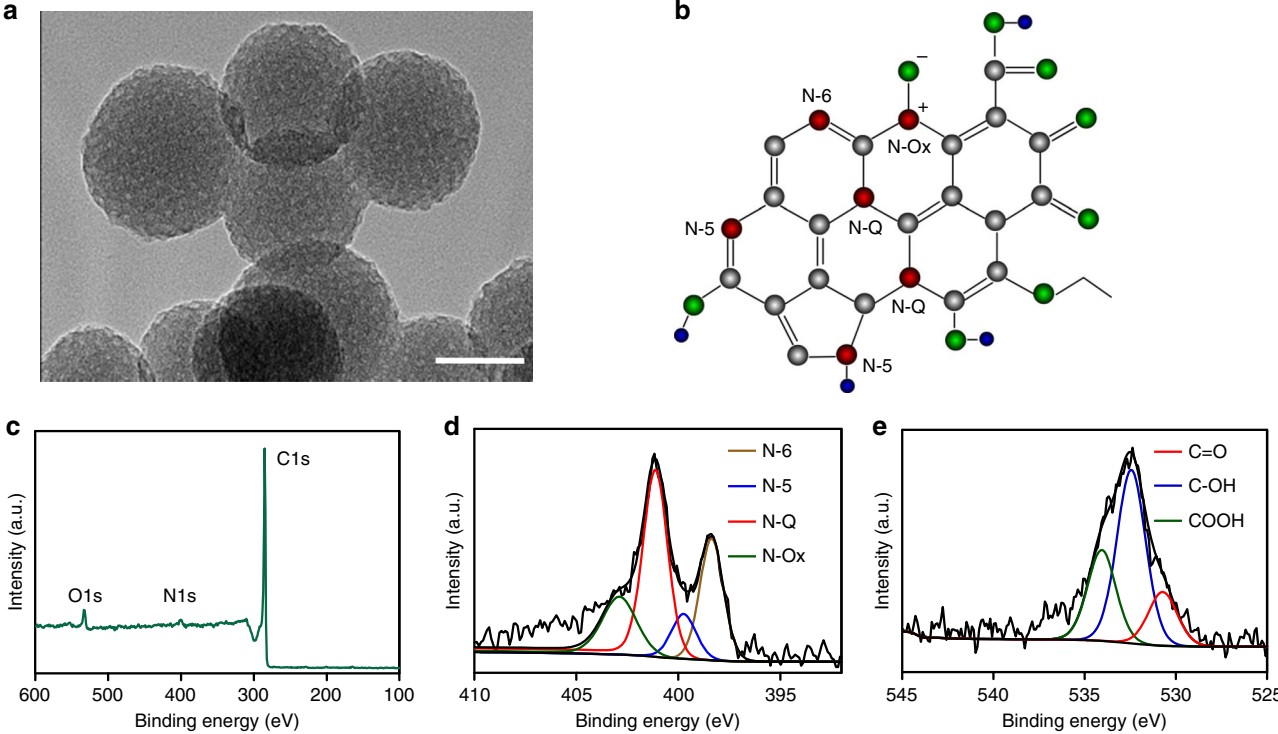

**Fig. 1** Morphology and N doping analysis of N-PCNSs. **a** HRTEM image of N-PCNSs-3. Scale bar: 100 nm. **b** Schematic model of nitrogen and oxygen containing surface functionality in N-PCNSs-3. **c** XPS survey spectra of N-PCNSs-3. **d** N 1s core-level XPS spectra of N-PCNSs-3. **e** O 1s core-level XPS spectra of N-PCNSs-3

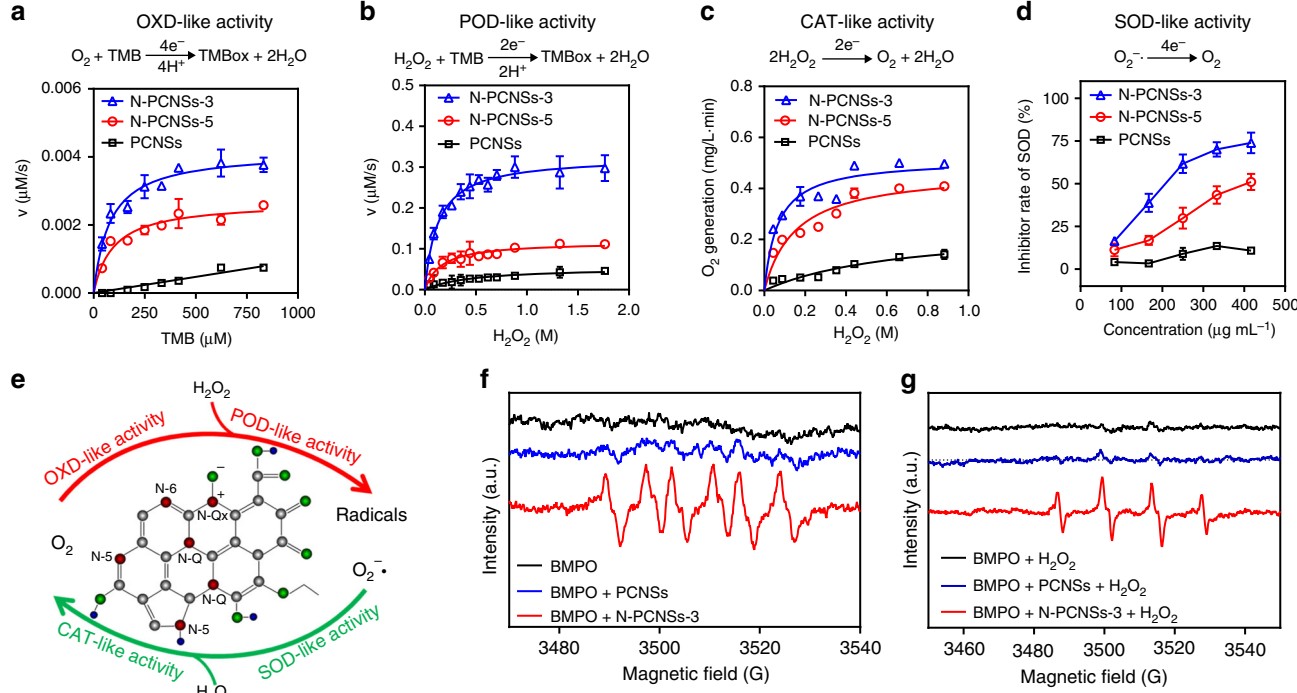

**Fig. 2** Enzyme kinetics of N-PCNSs. **a** Kinetics for OXD-like activity of N-PCNSs. **b** Kinetics for POD-like activity of N-PCNSs. **c** Kinetics for CAT-like activity of N-PCNSs. **d** SOD-like activity of N-PCNSs. **e** Schematic presentation of enzyme-like activities of N-PCNSs. **f**, **g** Generation of both superoxide radicals during OXD-like catalysis as well as hydroxyl radicals during POD-like catalysis. Error bars shown represent the standard error derived from three independent measurements

type of emerging nanozymes performing four enzyme-like activities which are able to catalyze ROS formation and scavenging. In particular, we develop an approach, using ferritin as a mediator, to specifically deliver N-PCNSs nanozymes into cell lysosome which allows the nanozymes to perform the desired activities responsible for ROS generation. The results reveal that N-PCNSs nanozymes, enabled by ferritin guidance, effectively suppress tumors in in vivo tests, thus indicating that tumor therapy can be reached by the coordinated catalysis of nanozymes.

## Results

**Characterization and enzyme-like activities of N-PCNSs.** To prepare N-doped porous carbon nanospheres (N-PCNSs), we used a pyrolysis method through a F127 soft template containing melamine and formalin/phenol as nitrogen and carbon sources, respectively[21–23]. The carbon nanospheres exhibited uniform size of approximately $100 \pm 10$ nm in diameter and porous structure on the surface characterized by transmission electron microscopy (TEM), scanning electron microscopy (SEM), and element mapping (Fig. 1a and Supplementary Figs. 1 and 2). The bonding configurations of the nitrogen atoms in N-PCNSs are shown in Fig. 1b. As shown in Fig. 1c, d, configuration of nitrogen was characterized by deconvolution N 1 s spectra. Analysis identified four typical peaks at 398.38, 399.78, 401.18, and 402.88 eV, representing pyridinic nitrogen (N-6), pyrrolic nitrogen (N-5), quaternary nitrogen (N-Q) and pyridine oxide or the oxidized nitrogen (N-O$_X$) groups. In order to study the role of N atoms in the catalytic process, we synthesized N-PCNSs with different N doping levels and a non-doped PCNSs as the control, which were termed as N-PCNSs-3 (high N), N-PCNSs-5 (low N) and PCNSs (without N), respectively (Supplementary Fig. 3). As shown in Supplementary Tables 1 and 2, the atom ratio of total nitrogen,

N-6 and N-Q in N-PCNSs-3 were higher than those in N-PCNSs-5, while that of N-5 was same. All other physicochemical characterizations, including X-ray diffraction (XRD), Raman spectroscopy and Fourier transform infrared spectroscopy (FTIR), nitrogen adsorption–desorption isotherm and corresponding discussions are shown in Supplementary Figs. 3-5, Supplementary Tables 1 and 2. Together, these results clearly indicated that nitrogen was successfully doped into the porous carbon framework of PCNSs.

Next, we tested whether N-PCNSs possess enzyme-like activities using the substrates for natural enzymes under physiological conditions. First, we investigated its oxidase (OXD)-like activity, which reduces oxygen into water in the presence of a hydrogen (H) donor (Fig. 2a). In order to monitor the reaction, 3,3′,5,5′-Tetramethylbenzidine (TMB) was introduced as the H donor, as the oxidized TMB develops a blue product with absorbance at 652 nm. N-PCNSs showed the capability to oxidize TMB in NaAc buffer (Supplementary Fig. 6a) in a pH-dependent and temperature-dependent manner, with maximum activity at pH 4.5 and temperature at 40 °C (Supplementary Figs. 6b and c). Both N-PCNSs-3 and N-PCNSs-5 exhibited Michaelis–Menten kinetics in the TMB colorimetric reaction. In comparison, PCNSs exhibited minimal activity (Fig. 2a). In addition, N-PCNSs-3 exhibited higher catalytic efficiency than N-PCNSs-5 as indicated by the values for $K_M$ and $V_{max}$ (Supplementary Table 3). Together, these results suggested that N dopant is critical for initiating OXD-like activity, and a higher concentration of N dopant increases the catalytic efficiency. The colorimetric reaction was further enhanced in an atmosphere with oxygen levels of 100%, and was inhibited in an oxygen-free $N_2$ atmosphere (Supplementary Fig. 6d), further confirming that the OXD-like activity represents a type of ORR.

We then tested peroxidase (POD)-like activity of N-PCNSs, which reduces hydrogen peroxide ($H_2O_2$), generating free radicals

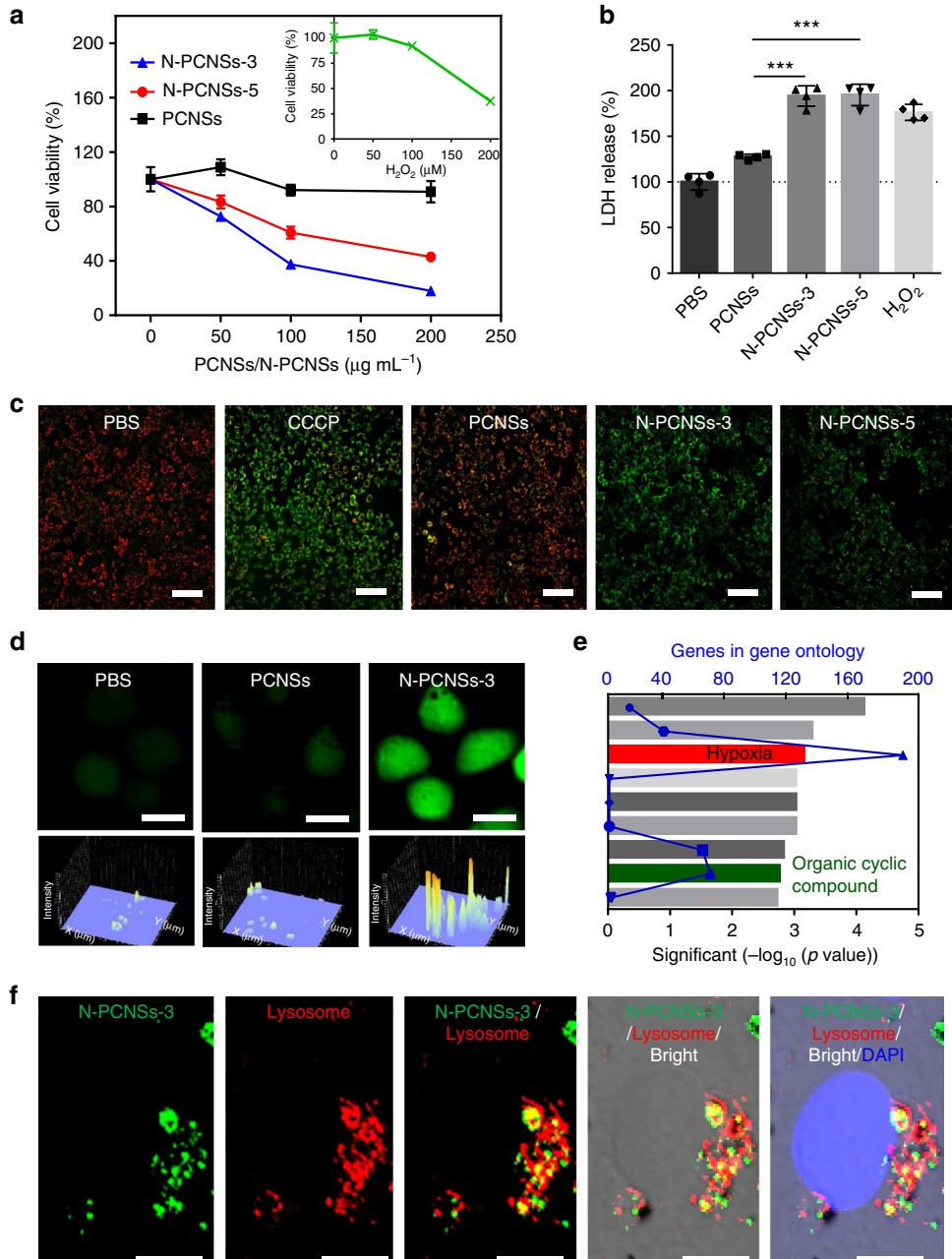

**Fig. 3** Cellular localization of N-PCNSs in HepG2 and its effects on viability. **a** Effect of N-PCNSs on HepG2 cell viability ($n = 6$). Insert: $H_2O_2$ cytotoxic effects on HepG2. **b** Determination of cell membrane integrity by LDH assay. Statistical significance is assessed by unpaired Student's two-sided $t$-test compared to the control group. ***$p < 0.001$. Mean values and error bars are defined as mean and s.d. ($n = 4$). **c** Influence of N-PCNSs treatment on mitochondrial membrane potential. Scale bar: 100 μm. **d** ROS level in HegG2 cells after N-PCNSs-3 stimulation. Scale bar: 20 μm. **e** Gene ontology analysis of biological process after N-PCNSs treatment. The changes of mRNA in responses to the hypoxia and organic cyclic compound were marked in the figure as Hypoxia and Organic cyclic compound, respectively. $p$ values are assessed by Fisher's exact test. **f** Localization of N-PCNSs in HepG2 cells. Scale bar: 10 μm. The Pearson's coefficient of N-PCNSs and lysosome is 0.8214

as a result (Fig. 2b). To test this activity, $H_2O_2$ was directly added into the colorimetric reaction together with TMB. As shown in Supplementary Fig. 7a, the reaction speed significantly increased in the presence of $H_2O_2$ in both the N-PCNSs and TMB system, relative to PCNSs which exhibited minimal activity. The activity was also pH dependent (pH 2.5–6.0 in 0.1 M NaAc buffer) with the optimal temperature at 40 °C (Supplementary Figs. 7b and c). Figure 2b and Supplementary Fig. 7d show the typical Michaelis–Menten kinetics for $H_2O_2$ and TMB substrates, respectively. N-PCNSs-3 exhibited the highest activity as

indicated by the values for $K_M$ and $V_{max}$ (Supplementary Table 4), suggesting that the affinity of PCNSs to the substrates significantly improved following N doping. More importantly, the $V_{max}$ of N-PCNSs-3 for TMB and $H_2O_2$ was improved up to 2.05 and 2.78-fold, compared to those of N-PCNSs-5 (Supplementary Table 4). Together, these results demonstrate that N doping is crucial for POD-like activity of $H_2O_2$ reduction.

In order to test if N-PCNSs possess catalase (CAT)-like activity, which splits $H_2O_2$ into $O_2$ and water (Supplementary Fig. 8a–c), we used a dissolved oxygen meter that measures

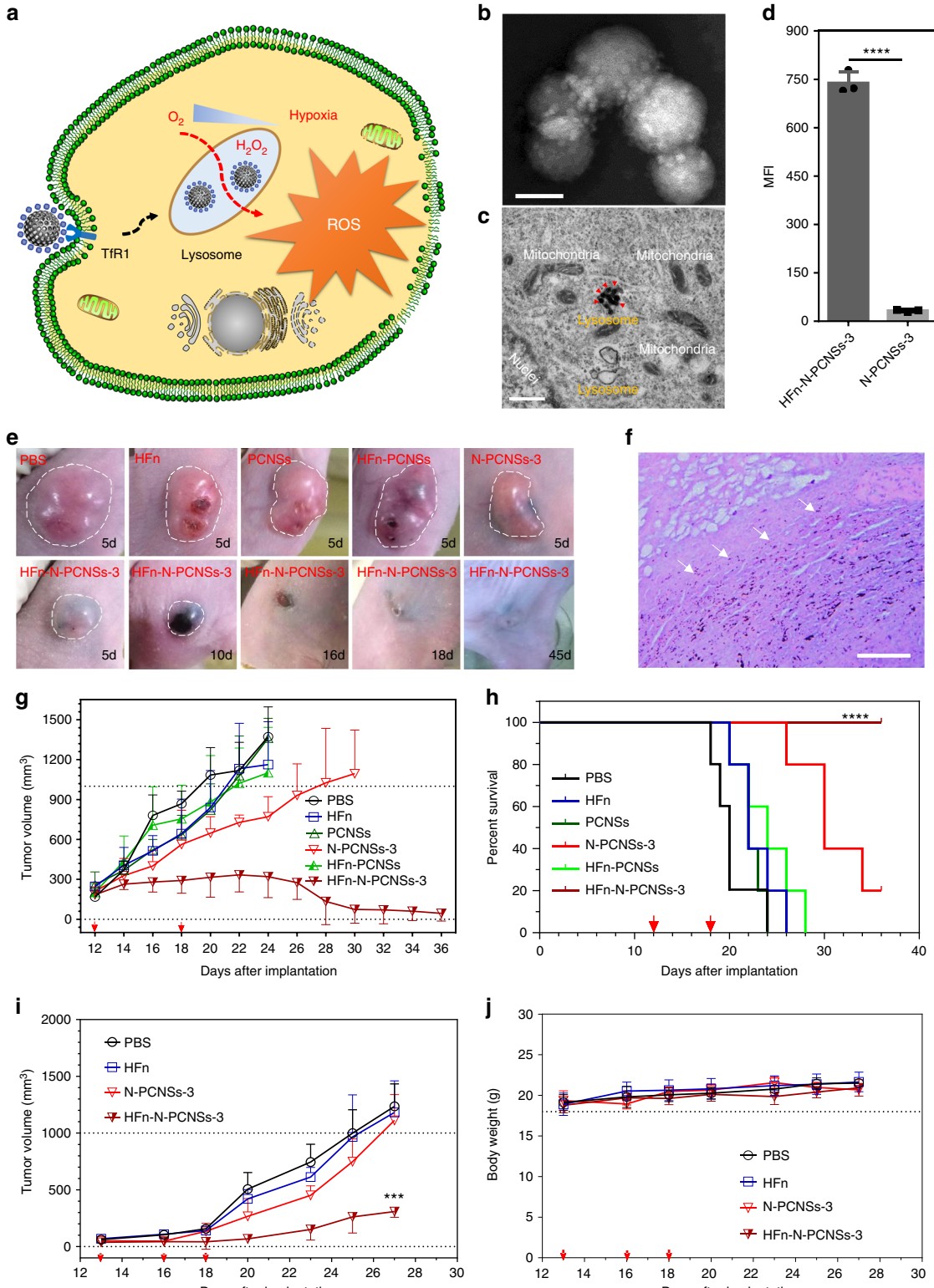

**Fig. 4** Tumor therapy based on N-PCNSs using HFn coordination. **a** Schematic for N-PCNSs induced tumor cell destruction via ferritin-mediated specific delivery. **b** TEM image of HFn assembled onto N-PCNSs. Scale bar: 50 nm. **c** TEM image of cancer cells treated with HFn-PCNSs. Red arrows indicate the position of HFn-PCNSs. Scale bar: 500 nm. **d** Quantification of HFn-enhanced internalization of N-PCNSs by flow cytometry analysis ($n = 3$). Statistical significance is assessed by unpaired Student's two-sided $t$-test compared to the control group. ****$p < 0.0001$. **e** Tumor morphology and progress with HFn-N-PCNSs treatment ($n = 5$). **f** Accumulation of N-PCNSs in tumors. White arrow: edge of the tumor area. Scale bar: 200 μm. **g** Tumor volume after *i.t.* treatment with HFn-N-PCNSs ($n = 5$). **h** Survival results of tumor therapy with HFn-N-PCNSs i.t. treatment ($n = 5$). Asterisks indicate ****$p < 0.0001$ for HFn-N-PCNSs-3 group compared with control groups (Kaplan–Meier). **i** Tumor volume after i.v. treatment with HFn-N-PCNSs ($n = 5$). Statistical significance is assessed by unpaired Student's two-sided $t$-test compared to the control groups. ***$p < 0.001$. **j** Body weight changes of treated mice following i.v. injection of HFn-N-PCNSs ($n = 5$). Mean values and error bars are defined as mean and s.d., respectively. i.t. intratumorally, i.v. intravenously

emerging oxygen. N-PCNSs exhibited CAT-like activity as determined by $O_2$ generation, with an optimal pH in the neutral range (pH 7). N-PCNSs-3 showed the highest activity when compared to N-PCNSs-5 and PCNSs. As shown in Fig. 2c, the decomposition of $H_2O_2$ by N-PCNSs followed a Michaelis–Menten kinetics. The $K_M$ of N-PCNSs-3 to $H_2O_2$ was 10 times lower than that of PCNSs and 2.3 lower than that of N-PCNSs-5 (Supplementary Table 5). The $V_{max}$ of N-PCNSs-3 showed 2.04 and 1.09-fold increases relative to PCNSs and N-PCNSs-5, respectively.

Lastly, we tested whether N-PCNSs possess superoxide dismutase (SOD)-like activity of N-PCNSs to reduce superoxide radical ($O_2^-\bullet$) which is more active than $H_2O_2$. As shown in Fig. 2d, at pH 8, both N-PCNSs-3 and N-PCNSs-5 possess SOD-like activity, with the former exhibiting a two-fold higher activity than the latter, indicating that the N dopant is critical for this catalytic activity. The CAT-like and SOD-like activities may follow the mechanism in OER process.

Taken together, these results demonstrated that N-PCNSs perform four enzyme-like activities under physiological conditions: OXD, POD, CAT and SOD (Fig. 2e). OXD-like and POD-like activities catalyze the reactions similar to oxygen reduction in ORR while CAT-like and SOD-like activities act on the reversible reactions similar to OER. According to the literature, heteroatomic nitrogen-doping is important for carbon nanomaterials to perform ORR and OER, however, it remains unclear which nitrogen type is critical. Yang et al. reported that the ORR is contributed by quaternary nitrogen and the OER closely related to pyridinic nitrogen under alkaline condition[24], but Guo's work revealed that pyridinic nitrogen play the key role for ORR in acidic electrolyte[19]. The N-PCNSs contain the above two nitrogen types and all enzymatic activities are enhanced when their ratios are increased. Although in-depth understanding is required to identify the roles of N types in the catalysis of enzyme-like activities, N doping is critical for N-PCNSs to perform these four activities. Besides N doping, the carbonization process involved in forming graphitic structure is also important for the enzyme-like activities. The precursor of N-PCNSs did not show any enzyme-like activities due to lacking carbonization to form proper N doping and graphitization (Supplementary Fig. 9). However, the substance with similar N types but no graphitic structure cannot perform the enzyme-like activities either. For instance, flavin adenine dinucleotide (FAD) which usually works as the cofactor in many enzymes, possesses pyridinic and quaternary nitrogen but cannot perform OXD-like, POD-like or CAT-like activities (Supplementary Fig. 10), indicating graphitic structure is required to conduct the enzyme-like activities. From these evidences, we conclude that both N doping and graphitic structure are requisite to achieve the above enzyme-like activities. On the basis of above analysis, we propose that N-PCNSs with N-dopant and graphitic structure can be used as peroxisomal mimetics that allow regulation of cellular ROS.

**ROS regulation in tumor cells**. To test the ability of N-PCNSs in ROS regulation, we first measured the in vitro generation of free radicals using electron spin resonance (ESR). As shown in Fig. 2f, g and Supplementary Fig. 11, free radicals of $O_2^-\bullet$ and OH• were observed and quenched by hypotaurine in the TMB colorimetric reaction, demonstrating that N-PCNSs possess the ability to generate ROS. To investigate intracellular ROS regulation, we used the human hepatocellular carcinoma cell line HepG2 as the model cell. As shown in Fig. 3a, both N-PCNSs-3 and N-PCNSs-5 inhibited HepG2 cell viability, with the toxic effects similar to those observed for $H_2O_2$.

In order to characterize the effects on cell membrane and mitochondria, we used a lactase dehydrogenase (LDH) quantification assay as well as fluorescence microscopy. As shown in Fig. 3b, release of LDH from the cytosol into the medium was increased 1.9-fold after N-PCNSs-3 treatment compared to the control (Fig. 3b), indicating that the cell membrane is damaged. The results of fluorescence imaging showed that the mitochondrial membrane potential (MMP) collapsed following addition of N-PCNSs-3 and N-PCNSs-5 (green color), while no such effects were observed upon incubation with PCNSs (red color; Fig. 3c), indicating that PCNSs trigger an apoptotic cascade inside the targeted cells after N doping. Together, these data strongly suggest that N-PCNSs-3 induces the greatest toxic effects to the cell and impaired cell viability, comparable to the effects of abnormally high ROS levels. To confirm the correlation between the toxic effects and ROS, we next quantified ROS levels in HepG2 cell after N-PCNSs-3 treatment. As shown in Fig. 3d and Supplementary Fig. 12, a marked increase in DCFH-DA staining was detected with N-PCNSs-3, which was up to 10-fold compared to levels measured following treatment with PCNSs, indicating that a ROS burst was initiated by the adding of N-PCNSs-3 to HepG2 cells. In addition, RNA-sequencing analysis showed that hypoxia was induced in HepG2 cells after N-PCNSs incubation, further reducing cell viability (Fig. 3e). These results suggest that N-PCNSs have the ability to induce an intracellular ROS burst that is accompanied by hypoxia, which in turn results in cell death.

The enzymatic analysis clearly shows that OXD-like and OPD-like activities of N-PCNSs generate ROS and CAT-like and SOD-like activities scavenge ROS. The above cell assays further showed that N-PCNSs resulted in higher ROS levels, not lower, indicating that the activities for ROS generation were favored in the cellular microenvironment. To address what favored the activities for ROS generation, we labeled N-PCNSs-3 with Alexa Fluor 488 to track their intracellular localization. As shown in Fig. 3f, we found that N-PCNSs-3 co-localized with late endosome/lysosomes, the highly acidic microenvironment of which would favor the ORR process with OXD and POD-like activities. In addition, we used IONPs (200 nm diameter) with POD-like and CAT-like activities as a control and found that these nanoparticles cannot induce cell toxicity or increase ROS level in HepG2 cells, indicating that the two activities alone are not able to induce sufficient impairment to the cells (Supplementary Figs. 13a and b)[25]. Together, these results provide strong evidence that N-PCNSs possess the ability to regulate intracellular ROS. Once located in the acidic microenvironment of lysosomes, they induce cell death through consuming oxygen to induce hypoxia and synergistically boosting the high level of ROS.

**Ferritin guiding N-PCNSs for in vivo tumor catalytic therapy**. It has been shown previously that stimulating ROS is a common strategy for cancer chemotherapy[26,27]. However, ROS also impairs normal cellular functions, thus proper ROS manipulation is essential for its use as an anti-tumor agent. It is critical to deliver ROS inducer specifically to tumor cells. For the N-PCNSs, they first must target tumor cells, and then also enter the lysosomal pathway for favoring ROS increase (Fig. 4a). In a recent report, we showed that hollow human H-ferritin nanoparticles (HFn lacking an iron core) specifically recognize tumor cells expressing high levels of HFn receptor (transferrin receptor 1, TfR1)[28] and thus deliver the complex into lysosomes via receptor-mediated endocytosis[29]. Therefore, we chose HFn to deliver the N-PCNSs into HepG2 cells as a Proof-of-Concept. We first conjugated HFn with N-PCNSs via activation of carboxyl groups of N-PCNSs (Fig. 1b, e) to react with amino-group of

HFn. TEM and Cryo-TEM were used to verify successful conjugation (Fig. 4b and Supplementary Fig. 14a). Figure 4c, d show that the efficiency of the internalization of HFn-N-PCNSs-3 and lysosome co-localization (Supplementary Fig. 14b) was approximately 23-fold higher than that for the un-conjugated control. More importantly, HFn-N-PCNSs-3 resulted in the same reduction in HepG2 cell viability as N-PCNSs-3 demonstrated (Supplementary Fig. 12c), but caused little effect on the non-tumor cell line hASMC (with low expression of TfR1, Supplementary Fig. 15). Together, these results indicate that HFn modification not only endows N-PCNSs with targeting ability to tumor cells, but also enables delivery of N-PCNSs to lysosomes where they initiate ROS production.

To evaluate the ability of HFn-N-PCNSs to target and destroy tumor cells in vivo, we employed an animal model using BALB/c nude mice bearing human HepG2 tumors. HFn-N-PCNSs-3 were intratumorally (i.t.) injected (at day 12 and 18 after tumor cell implantation), and the tumor progress was monitored using caliper measurements[30]. As shown in Fig. 4e, tumor growth was completely arrested within 5 days post treatment in the HFn-N-PCNSs-3 group. In comparison, the N-PCNSs-3 group only showed minor inhibitory effects, with HFn-PCNSs, HFn and PCNSs groups showed negligible effects on tumor growth. Furthermore, HFn-N-PCNSs-3 and HFn-PCNSs showed better uniform dispersal as determined by the black areas forming inside the tumor areas (Fig. 4e), indicating that HFn allows homogeneous delivery of PCNSs. The subsequent histochemical assessment of tumor sections showed that HFn-N-PCNSs-3 were primarily enriched in the tumor regions, with little signal detectable in peripheral tissues (Fig. 4f), providing further proof that HFn allows tumor-wide N-PCNSs delivery. We also found that tumor volumes decreased considerably, with survival rates significantly increasing after HFn-N-PCNSs-3 treatment (Fig. 4g, h and Supplementary Fig. 16a). In particular, two out of five tumors in this group completely disappeared during further observation of up to 45 days, with mortality reduced to zero in all mice treated with HFn-N-PCNSs-3. In addition, the body weight in the group treated with HFn-N-PCNSs-3 showed negligible change compared to the control group (Supplementary Fig. 16b). Histochemistry experiments showed that in HFn-N-PCNSs-3-treated group, healthy organs remained undamaged (Supplementary Fig. 17), indicating that i.t. administration of HFn-N-PCNSs-3 represents a safe approach for tumor therapy.

In order to evaluate the anti-tumor activity and the biosafety of HFn-N-PCNSs in a systemic manner, we intravenously (i.v.) injected HFn-N-PCNSs into human HepG2 tumor-bearing mice. As shown in Fig. 4i, HepG2 tumor growth was significantly inhibited in the HFn-N-PCNSs-3 group after 14 days post treatment. In comparison, the N-PCNSs-3 control group only exhibited minor inhibitory effects, with the HFn and PBS groups showing only negligible effects on tumor growth.

The systemic in vivo anti-tumor activity of HFn-N-PCNSs was also evaluated in human colon cancer HT-29 tumor xenograft mice model. The expression level of HFn receptor (TfR1) in HT-29 is comparable with that of HepG2 (Supplementary Fig. 18a). Both N-PCNSs-3 and HFn-PCNSs-3 inhibited HT-29 cell viability in vitro (Supplementary Fig. 18b). As shown in Supplementary Fig. 17c, in vivo HT-29 tumor growth was significantly inhibited in the HFn-N-PCNs-3 group. In comparison, the N-PCNSs-3, HFn and PBS groups showed minor or negligible effects on tumor growth.

The biosafety of HFn-N-PCNSs in a systemic manner was evaluated by systemic reaction (body weight changes) and pathological analysis. As shown in Fig. 4j and Supplementary Fig. 18d, no significant changes of body weight were detected between HFn-N-PCNSs-3 treatment group and control PBS

group in two different tumor models. Importantly, we demonstrated that the N-PCNSs are degraded under physiological conditions (Supplementary Fig. 19). In addition, our previous work demonstrated that most of the HFn nanocarriers accumulated in the healthy organs were eliminated from the body via kidney (into urine) and liver (into feces)[29]. The pathological analysis of the main organ tissues of healthy mice injected with HFn-N-PCNSs-3 indicated that HFn-N-PCNSs-3 failed to induce significant toxicity to these organs, including liver and kidney (Supplementary Fig. 20). These results indicate that i.v. administration of HFn-N-PCNSs-3 represents a safe approach for tumor therapy.

Together, these animal experiments demonstrate that with the guiding property of HFn, HFn-N-PCNSs-3 specifically bind to HepG2 or HT-29 tumor cells in vivo, then uniformly disperse in the tumor regions and shrink the tumor tissues, thus effectively treating the tumors and improving the survival rate of tumor-bearing mice. Moreover, because of the unique protein shell-cavity architecture of HFn, it is widely used as a nanocarrier for tumor-targeting drug delivery[31]. Importantly, recent findings demonstrated that chemotherapy drugs (e.g., doxorubicin, olaparib) induced/triggered the nuclear translocation of HFn, thus enhancing anti-tumor activity[32–34]. Taking advantage of the nanocarrier property of HFn should further improve the anti-tumor ability of this developed nanozyme catalytic tumor therapy strategy.

## Discussion

In summary, we designed a N-PCNSs system with four enzyme-like activities: OXD, POD, CAT and SOD, for oxygen oxidation and reduction under physiological conditions. Once localized to lysosomes, the multiple enzyme-like activities allow N-PCNSs to exert ROS mobilization including (1) transferring oxygen to free radicals accompanied with $O_2$ consumption, and (2) catalyzing $H_2O_2$ into free radicals, which thus synergistically bursts ROS levels and induce cell damage. Through the specific binding of ferritin with its receptor, N-PCNSs were successfully targeted to HFn receptor (TfR1)-positive tumors, localized to lysosome, ultimately resulting in tumor regression. These findings demonstrate that N-PCNSs are a type of emerging nanozyme that mimics multiple enzyme-like activities, and they are able to implement these catalysis in the compartment of a living cell.

In addition, heteroatomic doping to carbon nanomaterials is an effective approach to develop artificial enzymes (nanozymes). Although graphene and carbon nanotubes have been found with enzyme-like activity, most carbon nanomaterials exhibited no activity or weak activity of very limited catalytic types, such as POD-like or CAT-like activity[35–38]. N doping in carbon nanomaterials not only significantly improves the catalytic efficiency, but also allows them to perform multiple activities. Beside nitrogen, many other elements could be doped to the framework of carbon nanostructure[39]. Therefore, it is possible to make a series of carbon nanozymes with different specific activity or to mimic more enzyme-like activities. Importantly, it is necessary to determine the role of heteroatomic N in the catalysis, especially for the type of N corresponding to the types of enzyme-like activities. The role of graphitic structure also needs to be addressed to understand the catalytic mechanism. In this regard, N-PCNS is a good model of artificial enzyme to investigate the correlation between catalytic activities and nanostructure.

At the same time, it is critical to precisely control the in vivo destination and activity of the nanozyme, especially when a nanozyme possesses more than one activity under different reaction conditions. Right now there is still no proper way to control the performance of a nanozyme in a living cell. The

ferritin guiding system may be a universal platform to transport nanozymes to the acidic intracellular area. In our case the lysosome localization with acidic microenvironment allows N-PCNSs to perform OXD-like and POD-like activities, resulting in increasing ROS level to destruct tumor cells. The tumor therapy with N-PCNSs utilized the OXD-like and POD-like activities for ROS upregulation, while avoided CAT-like and SOD-like activities which scavenge ROS under neutral pH environment. Actually, it is possible to develop antioxidant therapy by latter two enzyme-like activities, which may be another potential application. In that case, the N-PCNSs also needs to be delivered to the right place in vivo.

Finally, the enzyme-like properties of N-doped carbon nanomaterials allow them to be used as a bioactive material, which will extend their application from chemical catalyst and energy study to biomedicine. The advantages of these carbon nanomaterials in multifunctionality, high stability and low cost will make them competent for in vivo catalysis, compared to natural enzymes which are easily denatured and degraded once entering circulation system. Taken together, we believe these catalytic carbon nanomaterials will have great potential for in vivo catalytic therapy in the future.

## Methods

**Materials**. Triblock copolymer Pluronic F127 (MW = 12600, $PEO_{106}PPO_{70}$-$PEO_{106}$) was purchased from Acros Organics (USA). Formalin aqueous solution (37 wt %), phenol, NaOH and melamine were purchased from Sinoparm Chemical reagent (China). 5-tert-butoxycarbonyl 5-methyl-1-pyrroline N-oxide (BMPO) was purchased from Radical Vision (Marseille, France). Flavin adenine dinucleotide (FAD) was purchased from J&K Scientific Ltd. (China). 1-ethyl-3-(3-dimethyl aminopropyl) carbodiimide (EDC) and N-hydroxysuccinimide (NHS). Hypotaurine, hemin, $H_2O_2$ (30%), 3,3',5,5'-tetramethylbenzidine (TMB), 1-ethyl-3-(3-dimethyl aminopropyl) carbodiimide (EDC) and N-hydroxysuccinimide (NHS), Triton X-100, paraformaldehyde, 2',7'-dichlorofluorescin diacetate ($H_2$DCFDA), carbonyl cyanide 3-chlorophenylhydrazone (CCCP) were purchased from Sigma-Aldrich; 4',6'-diamidino-2-phenylidole (DAPI) was purchased from Roche Applied Science. Poly-L-lysine-treated coverslips was purchased from BD Biosciences and six-well plate from Corning; Mouse anti-Lamp1 monoclonal antibody (clone H4A3) and goat anti-mouse IgG1 conjugated with Alexa-555 were purchased from Invitrogen. The human hepatocellular carcinoma cell line HepG2 were purchased from American Type Culture Collection (ATCC) and was cultured in DMEM medium (Sigma-Aldrich) containing 10% fetal calf serum (Sigma-Aldrich), penicillin (100 U mL$^{-1}$, Sigma-Aldrich) and streptomycin (100 g mL$^{-1}$, Sigma-Aldrich) at 37 °C with 5% $CO_2$. The human colon cancer cell line HT-29 was obtained from the ATCC and was cultured in RPMI-1640 medium (Sigma-Aldrich) containing 10% fetal calf serum (Sigma-Aldrich), penicillin (100 U mL$^{-1}$, Sigma-Aldrich) and streptomycin (100 g mL$^{-1}$, Sigma-Aldrich) at 37 °C with 5% $CO_2$. These cells are not listed by International Cell Line Authentication Committee as cross-contaminated or misidentified cell lines (v8.0, 2016). These cell lines have passed the conventional tests of cell line quality control methods (e.g., morphology, iso-enzymes, and mycoplasma).

**Synthesis of N-doped porous carbon nanospheres**. A synthesis procedure was applied to prepare N-doped porous carbon nanospheres (N-PCNSs) based on previous reports[21–23]. In details, 1.798 g melamine, 2.1 mL formalin aqueous solution (37.0 wt %) and 15 mL distilled water were mixed in a three-necked flask and stirred at 80 °C to obtain colorless transparent solution. Subsequently, 0.6 g phenol, 2.1 mL formalin aqueous solution (37.0 wt %) and 15 mL NaOH (0.1 mol L$^{-1}$) aqueous solution were added to the solution and stirred at 66 °C for an additional 30 min to obtain low-molecular-weight phenolic resols. In sequence, 15 mL Pluronic F127 (0.7 g) aqueous solution was added, and then the mixture was stirred for 2 h at 66 °C with a stirring speed of 350 r min$^{-1}$. Next, 50 mL water was added to dilute the solution. During this reaction, the color of the solution turned from colorless transparent to pink and finally to crimson. 20 h later, the reaction was stopped and the precipitate was observed. Once the precipitate was dissolved, 17.7 mL of the solution was heated for 24 h at 130 °C in an autoclave. Finally, the products were collected by centrifugation (12,000×g, 15 min), washed with water several times and dried at room temperature. The dried powder was carbonized at 800 °C in a $N_2$ atmosphere to remove the Pluronic F127 templates, and the resulting N-PCNSs (named N-PCNSs-3 for 3 h carbonization, N-PCNSs-5 for 5 h carbonization) were then stored at room temperature for further use. Non-N-doped PCNSs were used as a control, and were synthesized according to the identical experimental procedures, without addition of melamine in the initial process.

**Characterization of N-PCNSs**. The morphology of porous carbon nanospheres was characterized by transmission electron microscopy (TEM) (Tecnai 12, Philip, Netherlands) and high-resolution transmission electron microscopy (HR-TEM) (Tecnai G2 F30 S-TWIN, FEI, USA). For TEM images, the samples diluted in ethanol were dropped onto a copper grid. The ultraviolet-visible (UV-Vis) absorption spectra and the time-dependent absorbance spectra were measured with a Lambda 650 S Spectrophotometer (PerkinElmer, USA). X-ray Photoelectron Spectroscopy (XPS) was used for the characterization of chemical composition and state of elements present in the investigated porous carbon nanospheres. XPS spectra were recorded on a Thermo ESCALAB 250 spectrometer using an Al Kα X-ray source. The Raman spectra were collected at room temperature using a Renishaw inVia spectrophotometer. The crystalline structures of the as prepared samples were evaluated by X-ray diffraction (XRD) analysis on a D8 ADVANCE diffractometer by using Cu Kα radiation. Electron spin resonance measurement was carried out using a Bruker ESR spectrometer (A300-10/12, Germany) at ambient temperature. Fourier transform infrared (FTIR) spectra were performed using a IRAffinity-1 Spectrometer (Shimadzu, Japan). Nitrogen adsorption–desorption isotherms were performed on a surface area and pore size analyzer (Micromeritics ASAP 2020 HD88, USA).

**Oxidase-like activity of N-PCNSs and kinetic assay**. The oxidation of TMB by N-PCNSs in HAc-NaAc buffer produced a blue signal with major absorbance peaks at 370 and 652 nm. In a typical test, chemicals were added into 1.0 mL buffer solution (0.1 M HAc-NaAc buffer, pH 4.5) in an order of certain amounts of the N-PCNSs and 10 μL TMB (final concentration 0.416 mM). Kinetic measurements of the oxidase reactions of N-PCNSs were performed using a Lambda 650SUV-Vis spectrophotometer at 652 nm. The kinetic assays of N-PCNSs with TMB as the substrate were performed by adding different amounts (1, 2, 4, 6, 8, 10, 12.5, 15, 20 μL) of TMB solution (in DMSO, 10 mg mL$^{-1}$). The Michaelis–Menten constant was calculated according to the Michaelis–Menten saturation curve by GraphPad Prism 6.02 (GraphPad Software). For comparison, the oxidase-like activity of PCNSs was also measured.

**Peroxidase-like activity of N-PCNSs and kinetic assay**. The peroxidase (POD)-like activity assays of N-PCNSs were carried out using TMB as the substrate in the presence of $H_2O_2$ in 0.1 M HAc-NaAc buffer solution (pH 4.5). The absorbance of the color reaction (at 652 nm for TMB) was recorded at a certain reaction time via a Lambda 650 S UV-Vis spectrophotometer to express the POD-like activity. Typically, chemicals were added into 1.0 mL buffer solution in an order of certain amounts of the N-PCNSs, 10 μL TMB (final concentration 0.416 mM), and 60 μL $H_2O_2$ (final concentration 0.5292 M) to show the chromogenic reactions implying POD-like activity. The steady-state kinetic assays were conducted at 40 °C in a 1 mL reaction buffer solution (0.1 M HAc-NaAc, pH 4.5) with 10 μL N-PCNSs solution (2.5 mg mL$^{-1}$) as catalyst in the presence of $H_2O_2$ and TMB. The kinetic assays of N-PCNSs with TMB as the substrate were performed by adding 10 μL 30% $H_2O_2$ and different amounts (1, 2, 4, 6, 8, 10, 12.5, 15, 17.5, 20 μL) of TMB solution (in DMSO, 10 mg mL$^{-1}$). The kinetic assays of N-PCNSs with $H_2O_2$ as the substrate were performed by adding 10 μL of 10 mg mL$^{-1}$ TMB and different amounts (5, 10, 20, 30, 40, 50, 60, 70, 80, 100, 150, 200 μL) of 30% $H_2O_2$ solution. All reactions were monitored by measuring the absorbance at different reaction times, and the Michaelis–Menten constant was calculated according to the Michaelis–Menten saturation curve by GraphPad Prism 6.02 (GraphPad Software). The pH dependence of the POD-like activity of N-PCNSs was detected in different buffer solutions with pH values from 2 to 11, and the temperature dependence was detected in different temperatures increased gradually from 25 °C to 60 °C. For comparison, the POD-like activity of PCNSs was also measured.

**Catalase-like activity of N-PCNSs and kinetic assay**. Catalase (CAT)-like activity assays of N-PCNSs were carried out at room temperature by measuring the generated oxygen using a specific oxygen electrode on Multi-Parameter Analyzer (JPSJ-606L, Leici China). In a typical test, chemicals were added into 3.0 mL buffer solution (0.1 M PB buffer, pH 7.0) in the order of 0.2 mL (2.5 mg mL$^{-1}$) N-PCNSs and then 180 μL 30% $H_2O_2$ solution. The generated $O_2$ solubility (unit: mg L$^{-1}$) was measured at different reaction times. The kinetic assays of N-PCNSs with $H_2O_2$ as the substrate were performed by adding different amounts (0, 15, 30, 60, 90, 120, 150, 225, 300 μL) of 30% $H_2O_2$ solution. The Michaelis–Menten constant was calculated according to the Michaelis–Menten saturation curve by GraphPad Prism 6.02 (GraphPad Software). The pH dependence of the CAT-like activity of N-PCNSs was identified in different buffer solutions with pH values from 2 to 11. For comparison, the CAT-like activity of PCNSs was also conducted.

**SOD-like activity of N-PCNSs**. The SOD-like activity assays of N-PCNSs were carried out at room temperature by employing a commercial colorimetric SOD assay kit (S311-10) from Dojindo Molecular Technologies. The assay was performed according to the manufacturer's instructions, and the SOD-like activity of a series concentrations of N-PCNSs was expressed as the percentage inhibition of the competitive WST reaction with superoxide by natural SOD enzyme.

**Free radical identification**. All ESR measurements were carried out using a Bruker electron spin resonance (ESR) spectrometer (A300-10/12, Germany) at ambient temperature. Fifty microliter aliquots of control or sample solutions were put in glass capillary tubes with internal diameters of 1 mm and sealed. The capillary tubes were inserted into the ESR cavity, and the spectra were recorded at selected times. Instrument settings were as follows: 1 G field modulation, 100 G scan range, and 20 mW microwave power for detection of spin adducts using spin traps. The spin trap BMPO was employed to verify the formation of hydroxyl radicals (OH•) during the degradation of $H_2O_2$ in presence of PCNSs or N-PCNSs-3 under various conditions. The amount of hydroxyl radicals was quantitatively estimated by the ESR signal intensity of the hydroxyl radical spin adduct (BMPO/OH•) using the peak-to-peak height of the second line of the ESR spectrum. To verify the superoxide anion ($O_2^-$•), BMPO was also used to trap the superoxide in the form of spin adduct BMPO/$O_2^-$•.

To investigate the effect of hypotaurine on ROS produced by N-PCNSs-3, we incubated TMB and N-PCNSs-3 in 1.0 mL NaAc-HAc buffer (0.1 M, pH 4.5) containing 4% hypotaurine with or without $H_2O_2$. The absorbance of TMB at 652 nm was recorded to evaluate the effect of hypotaurine inhibition.

**Cell viability assay and transcriptome analysis**. In vitro cytotoxicity of N-PCNSs was determined using the CCK-8 cell viability kit assay (Dojindo Molecular Technologies). Briefly, HepG2 cells were plated in 96-well plates (BD Biosciences) with a density of $5 \times 10^3$ cells per well and cultured in 100 μL RPMI 1640 medium for 1 day before addition of N-PCNSs or $H_2O_2$. On each plate, blank wells ($n = 6$) with media were defined as 0%. And the wells with PBS-treated cells only ($n = 6$) were defined as 100% viability. Dilutions of N-PCNSs were prepared in PBS. The cells were then exposed to a series of concentrations (0 to 200 μg mL$^{-1}$) of N-PCNSs or (0 to 200 μM) $H_2O_2$ for 48 h. After stimulation, 10 μL CCK-8 solution was added to each well. The plates were incubated for 4 h at 37 °C. The absorbance was then determined at 450 nm using a Benchmark Plus microplate spectrophotometer (Bio-Rad Laboratories, Inc.). The results presented are the average of three independent experiments.

Transcriptome analysis of N-PCNSs and PBS-treated HepG2 cells, including total RNA extracting, RNA-sequencing and bioinformatic data analysis were performed by Shanghai Novelbio Ltd.

**LDH assay**. Cell membrane integrity was determined by measuring LDH activity. After incubation with N-PCNSs (200 μg mL$^{-1}$) or (200 μM) $H_2O_2$ for 48 h, the supernatant of HepG2 was collected, and the amount of LDH released from cells was determined using LDH assay kit (QuantiChrom$^{TM}$ LDH cytotoxicity assay kit, BioAssay Systems, USA) according to the manufacturer's instructions. The absorbance was measured on a Benchmark Plus microplate spectrophotometer (Bio-Rad Laboratories, Inc.) at 490 nm. The data in each treatment group is expressed as a percentage of control (PBS treatment). The results presented are the average of three independent experiments.

**Mitochondria potential assay**. Loss of mitochondrial membrane potential was assessed by confocal laser scanning microscope using the dye JC-1 (JC-1 assay kit, Beyotime, China). Briefly, HepG2 cells were plated on poly-L-lysine-treated coverslips (BD Biosciences) and cultured in a six-well plate (Corning) for 12 h before using. After treatment with 200 μg mL$^{-1}$ N-PCNSs for 48 h, HepG2 cells were stained with JC-1 for 20 min at 37 °C. After washed twice following the manufacturer's instructions, cells on slides were scanned with a confocal laser scanning microscope (Olympus FluoView FV-1000, Tokyo, Japan). Red emission of the dye represented a potential-dependent aggregation in the mitochondria. Green fluorescence represented the monomeric form of JC-1, appearing in the cytosol after depolarization of the mitochondrial membrane. Cells treated with 10 mM CCCP (carbonyl cyanide 3-chlorophenylhydrazone) were used as a positive control.

**Cellular ROS assay**. The fluorescent probe 2′,7′-dichlorofluorescin diacetate (H₂DCFDA, Sigma-Aldrich, D6883) was used to measure the intracellular generation of ROS by N-PCNSs. Briefly, confluent HepG2 cells on coverslips (BD Biosciences) were incubated with 200 μg mL$^{-1}$ N-PCNSs for 48 h. After washing with PBS, the cells were incubated with 10 μM H₂DCFDA in serum-free DMEM for 20 min at 37 °C in the dark. The fluorescence intensities of H₂DCFDA were measured by confocal laser scanning microscope (Olympus FluoView FV-1000, Tokyo, Japan). The quantitative analysis of the intensity of H₂DCFDA was performed using Olympus FluoView Ver.1.7a Viewer. For flow cytometry analysis of ROS, after incubation with H₂DCFDA, cells were measured immediately by flow cytometry (FACSCalibur$^{TM}$, Becton Dickinson, Franklin Lakes, NJ, USA) with excitation at 488 nm and emission at 530 nm. Green mean fluorescence intensities were analyzed using FlowJo7.6 software (Tree Star, OR, USA).

**Localization in cytoplasm**. The cellular uptake and distribution of N-PCNSs in cells was investigated by confocal laser scanning microscope. Briefly, HepG2 cells were plated on poly-L-lysine-treated coverslips (BD Biosciences) and cultured in a six-well plates (Corning) for 12 h before use. After stimulation for 48 h with 200 μg mL$^{-1}$ Alexa-488 labeled N-PCNSs, the cells were washed with PBS, fixed in 4% cold formaldehyde in PBS for 5 min, and then permeabilized with 0.1% Triton X-

100. After washing with PBS, the cells were blocked in 5% normal goat serum for 30 min at room temperature. To visualize lysosomes, cells were incubated with an anti-Lamp1 mAb (1:200, clone H4A3; Invitrogen) at 37 °C for 1 h. Cells were then washed three times with PBS and incubated with goat anti-mouse IgG1 conjugated with Alexa-555 (1:500; Catalog: A32727, Invitrogen) for 1 h at 37 °C. Finally, the nuclei of cells were stained with 4′, 6′-diamidino-2-phenylidole (DAPI, 1 μg mL$^{-1}$, Roche Applied Science) for 10 min at room temperature. The samples were examined with a confocal laser scanning microscope (Olympus FluoView FV-1000, Tokyo, Japan).

**Ferritin conjugation and characterization on N-PCNSs**. Covalent attachment of HFn to N-PCNSs was performed using 1-ethyl-3-(3-dimethyl aminopropyl) carbodiimide (EDC) and N-hydroxysuccinimide (NHS). In a typical experiment, 5 mg N-PCNSs were incubated in 1.5 ml PBS containing 0.27 mg mL$^{-1}$ EDC, 0.4 mg mL$^{-1}$ NHS for 25 min, and then the N-PCNSs were centrifuged (12,000×g, 15 min) and washed with PBS twice. Next, 2 mg mL$^{-1}$ HFn was added into the above system. After reaction for 2 h, the solution was centrifuged (12,000×g, 15 min) and washed with PBS twice. Finally, HFn-N-PCNSs were blocked with BSA (5 mg mL$^{-1}$) for 1 h, followed by washing with PBS and redispersed in PBS, and the final product was kept at 4 °C for further use. The content of HFn conjugation was calculated by the BCA method.

**TEM and Cyro-EM analysis of HFn-N-PCNSs**. HFn-N-PCNSs solution with a concentration of 0.4 mg mL$^{-1}$ was pipetted/placed onto a 200-mesh copper grid. After staining with 2% (w/v) phosphomolybdic acid, the morphology of HFn-N-PCNSs liposomes was characterized by TEM. For Cryo-TEM observation, the HFn-N-PCNSs samples (3 μL, 0.25 mg mL$^{-1}$) were embedded in vitreous ice using an FEI Vitrobot Mark VI (FEI, Oregon) and images taken using an FEI 300-kV Titan Krios Cryo-TEM (FEI, Oregon) equipped with a Falcon3 camera. HFn-N-PCNSs were photographed in normal mode of the Falcon3 with a pixel size of 1.76 Å, and the total dose for each micrograph was about 40 e-/Å.

**Ferritin-mediated internalization and localization in cells**. Cellular uptake and distribution of HFn modified N-PCNSs in cells was studied using confocal laser scanning microscopy. Briefly, HepG2 cells were plated on poly-L-lysine-treated coverslips (BD Biosciences) and cultured in a six-well plate (Corning) until confluent. After stimulation for 48 h with 200 μg mL$^{-1}$ HFn-N-PCNSs or N-PCNSs, the cells were washed with PBS, fixed in 4% cold formaldehyde in PBS for 5 min, and then permeabilized with 0.1% Triton X-100. After washing with PBS, the cells were blocked with 5% normal goat serum for 30 min at room temperature. To visualize lysosomes, cells were incubated with an anti-Lamp1 mAb (1:200, clone H4A3; Invitrogen) at 37 °C for 1 h. Cells were then washed three times with PBS and incubated with goat anti-mouse IgG1 conjugated with Alexa-555 (1:500; Catalog: A32727, Invitrogen) for 1 h at 37 °C. Finally, the nuclei of cells were stained with 4′, 6′-diamidino-2-phenylidole (DAPI, 1 μg mL$^{-1}$, Roche Applied Science) for 10 min at room temperature. The coverslips were examined with a CLSM (Olympus FluoView FV-1000, Tokyo, Japan). The HFn-N-PCNSs and N-PCNSs were directly observed by bright view of CLSM. The areas of HFn-PCNSs and N-PCNs in CLSM images were quantitatively analyzed by Image J (NIH, USA).

**Western blotting**. TfR1 expression was assessed by Western blotting. Lysates of each cell type were run on 10% SDS-polyacrylamide gels and transferred onto nitrocellulose membranes, which were blocked with 5% non-fat milk/0.1% Tween-20 in phosphate-buffered saline (PBS; 30 min) and then incubated (overnight, 4 °C) with rabbit anti-human TfR1 polyclonal antibody (1:2000; Catalog: HPA028598, Sigma). Immunoreacted bands were detected using HRP-conjugated goat anti-rabbit IgG (1:6000; Catalog: 65–6120, Invitrogen), and developed using an ECL substrate (Pierce). The internal control was β-actin.

**Tumor therapy with HFn-N-PCNSs**. All animal studies were performed following the protocol approved by the Institutional Animal Care and Use Committee of the Institute of Biophysics, Chinese Academy of Sciences. Six to eight-week-old female BALB/c mice and BALB/c nude mice were obtained from the Vital River Laboratories (Beijing). Mice were housed under standard conditions with free access to sterile food and water. To establish the HepG2 or HT-29 xenograft tumor model, mice were injected subcutaneously with $2 \times 10^6$ HepG2 cells or HT-29 in 200 μL PBS.

For intratumoral injection studies, 30 female BALB/c nude mice bearing HepG2 tumors were randomly assigned to six groups ($n = 5$ mice per group) when the size of tumor reached about 200 mm³ (at 12 d after tumor implantation). The tumor volume was calculated according to the formula: V = 1/2 a × b², where a represents the tumor length, and b represents the tumor width. For carbon materials treatment groups, including PCNSs, N-PCNSs-3, HFn-PCNSs and HFn-N-PCNSs-3, 150 μL of carbon solution (2 mg mL$^{-1}$) was intratumorally (i.t.) injected into HepG2 tumor-bearing mice on day 12 and day 18 after tumor implantation. The mice treated with PBS or HFn protein with equivalent protein concentration with HFn-PCNSs and HFn-N-PCNSs-3 was used as negative controls.

For intravenous injection studies, 20 female BALB/c nude mice bearing HepG2 or HT-29 tumors were randomly assigned to four groups ($n = 5$ mice per group) when the size of tumor reached about 100 mm[3]. For carbon materials treatment groups, including N-PCNSs-3 and HFn-N-PCNSs-3, 200 μL of carbon solution (2 mg mL$^{-1}$) was intravenously (i.v.) injected via tail vain on day 13, 16 and day 18 after tumor implantation for HepG2 tumor-bearing mice, or on day 12, 15 and day 17 for HT-29 tumor-bearing mice. Mice treated with PBS or HFn protein with equivalent protein concentration with HFn-N-PCNSs-3 were used as negative controls.

The tumor size and body weight were measured 3 times a week. Measured values were presented as mean ± s.d. During treatment, mice were monitored for body weight loss and euthanized upon exceeding 15% loss in body weight or if the volume of tumors was more than 1000 mm$^3$. After therapy, major organs as well as tumors were collected, fixed in 4% paraformaldehyde, embedded in paraffin, sectioned at 5 μm, and the pathological examination was performed by hematoxylin eosin staining. Cumulative survival curves were compared using Kaplan–Meier analysis and the log-rank test using GraphPad Prism 6.02 (GraphPad Software).

For biosafety analysis, 15 female BALB/c mice were randomly assigned to 3 groups ($n = 5$ mice per group). Identical doses of HFn-N-PCNSs-3 materials as intravenous injection therapy studies were i.v. injected to mice. Mice treated with PBS were used as negative control. Mice were sacrificed on day 6 and day 30 post HFn-N-PCNSs-3 injection. Major organs were first fixed in 4% paraformaldehyde, then embedded in paraffin and sectioned at 5 μm, before being stained with hematoxylin eosin.

**Statistic methods**. The significance of the data in Figs. 3b and 4d, i is analyzed according to unpaired Student's two-sided $t$-test: ***$p < 0.001$. The $p$ values in Fig. 3e are assessed by Fisher's exact test. Cumulative survival curves in Fig. 4 were compared using Kaplan–Meier analysis and the log-rank test by GraphPad Prism 6.02 (GraphPad Software). The samples/animals were allocated to experimental groups and processed randomly.

**Data availability**. All data are available from the authors upon reasonable request.

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

## Acknowledgements

We thank the Testing and Analysis Centre at Yangzhou University for nanomaterial characterizations. We thank Dr. Xiaojun Huang, Dr. Gang Ji and Lei Sun for their excellent technical support in Cryo-EM and TEM imaging at the Transmission EM Facilities, Center for Biological Imaging, Institute of Biophysics. We thank Dr. Guizhi Shi, Ms Di Liu (Institute of Biophysics, Chinese Academy of Sciences (CAS)) for assistance with pathological analysis. We thank Mr. Ruofei Zhang and Mr. Bing Jiang (Institute of Biophysics, CAS) for their assistant with animal experiments. This work was supported by the Key Research Program of Frontier Sciences, CAS (Grant No. QYZDB-SSW-SMC013), National Key R&D Program of China (2017YFA0205501), the National Natural Science Foundation of China (Grant No. 31530026 and 81671810), the Strategic Priority Research Program, CAS (Grant No. XDA09030306, XDPB0304), the Foundation of the Thousand Talents Plan for Young Professionals and Jiangsu Specially-Appointed Professor, the Interdisciplinary Funding at Yangzhou University, Young Elite Scientist

Sponsorship Program by CAST (2015QNRC001), China Postdoctoral Science Foundation (Grant No. 2015M570158) and the China Postdoctoral Science Special Foundation (Grant No. 2016T90143).

## Author contributions

L.G., X. Y., and K.F. conceived and designed the experiments with J.X. and L.F. J.X., L.F., Y.T. and X.X. synthesized and characterized the materials. J.X., L.G. and C.Z. contributed to enzyme kinetics assay and protein conjugation. K.F., P.W. and B.J. performed cell and animal experiments. L.G., X.Y., K.F. and J.X. analyzed the data and wrote the paper. All authors discussed the results and commented on the manuscript.

## Additional information

**Competing interests:** The authors declare no competing interests.

