## [Peer Review File · Nature Communications]

Reviewers' comments:

Reviewer #1 (Remarks to the Author):

The paper of Kelong Fan and colleagues reported the development of a novel nanozyme using nitrogen-doped carbon nanospheres. This nanozyme displays four enzyme-like activities acting as oxidase, peroxidase, catalase and superoxide dismutase and it seems to have a good activity on Reactive Oxygen Species regulation. The nanozyme has been decorated with apoferritin to allow TfR1 mediated uptake and to obtain tumor targeted internalization. Therefore, it has been suggested as anti-cancer agent. The work is nice, well-written and interesting for a broad audience. Nanozyme activities are well-characterized and the idea to target in tumor cells nanozymes using apoferritin is novel. However, in reviewer's opinion *in vivo* experiments in cell culture and murine model should be refined to make this nanozyme really convincing for clinical applications. Indeed, administration should be performed by systemic route in order to evaluate also nanozyme bioavailability, tumor targeting capability, biodistribution in off-target organs and toxicity, which are not considered in this work but are issues to assess if we want to propose new strategies to treat cancer. Therefore, I suggest to reconsider this paper after major revisions.

In detail:

- 1) Figure 3 f: To correctly study colocalization between N-PCNs-3 and Lysosomes a Pearson's coefficient or other correlation studies should be provided. Moreover LAMP-1 marker is not a Lysosome specific marker since it could be found also in late endosomes (Phillip A. Vanlandingham and Brian P. Ceresa, *J Biol Chem.* 2009; 284(18): 12110–12124).
- 2) What kind of ferritin is used in this paper? Some works reported the absence of colocalization between H chain ferritin and lysosomes, using Cathepsin D as Lysosome marker. Moreover, they reported that H-chain ferritin as good vector for nuclear delivery (Zhang, L., et al. (2015), *Adv. Healthcare Mater.*, 4: 1305–1310. doi:10.1002/adhm.201500226; Mazzucchelli, S., et al. (2017). *Scientific Reports*, 7(1), 7505. DOI: 10.1038/s41598-017-07617-7; Mazzucchelli S. et al. *Oncotarget*. 2017 Jan 31;8(5):8383-8396. doi: 0.18632/oncotarget.14204.; Bellini M, et al. *J Control Release*. 2014 Dec 28;196:184-96. doi: 10.1016/j.jconrel.2014.10.002. I think you should specify what kind of ferritin you have used here and eventually improve the discussion about lysosomal localization of the Ferritin-decorated nanozyme.
- 3) Figure 4 c: It is not possible to study colocalization between a fluorescence signal acquired in confocal microscopy and the nanozyme signal acquired in bright field since the last one has not an adequate spatial definition along the z-axis.
- 4) Figure 4e-f-g-h: The displayed results are very interesting but have been obtained with intratumoral injection of the Nanozymes. I think this is the main bias of the work. In this way you demonstrate the efficacy on the tumor but you cannot demonstrate "safety" for healthy cells and tumor targeting capability *in vivo*. It should be very nice to see the same results obtained after an *i.v.* administration in order to evaluate also bioavailability and biodistribution. Moreover, the *in vivo* efficacy should be confirmed using other tumor models.
- 5) Supp. Figure 10. The demonstration of tumor targeted internalization by bright field microscope images is not convincing. You could perform an uptake study by flow cytometry using fluorescent labelled ferritin-decorated nanozymes or a confocal microscopy analysis.

Minor remark:

- 1) Figure 3 c: What is CCCP? I suppose it is the positive control. I think it should be better to increase figure readability describing the meaning of green and red spots.
- 2) Line 156-157: "indicating that PCNs trigger an apoptotic cascade inside the targeted cells only after N-doping. The use of the term "only" is not appropriate: there are green spots also in PCNs sample.
- 3) Line 227: "...healthy organs remained undamaged..." Of course, you have not performed a systemic administration! To assess systemic toxicity you should perform a systemic administration.
- 4) Methods section: western blot analysis reported in Supp. Fig. 11 is not described.

5) Supp. Figure 11. A viability assay on healthy Tfr1 expressing cell line should be performed to clarify what happens in healthy cells with high Tfr1 expression such as endothelial cells.

Reviewer #2 (Remarks to the Author):

Recommendation or Remarks: Major Revisions

Comments to Authors

This manuscript authored by Yan, Gao and co-workers describes the potential of N-doped carbon nanospheres (N-PCNs), in particular, N-PCNs-3 as four-in-one nanozyme for in vivo tumor catalytic therapy. Given the interesting as well as surprising features of N-PCNs (in terms of exhibiting four different enzyme-like activities), indubitably, this is an excellent research work in the fields of bioanalytical and biomedical sciences as well as carbon-based nanomaterials (NMs). However, in view of several open questions in conjunction with inadequate characterization of reported material(s), insufficient and/or inappropriate citation(s), and contemptible text (at many instances), I personally feel that the manuscript is not suitable for publication in its present form. Consequently, I recommend its publication in Nature Communications subjected to major revisions. For authors' reference, some of my remarks/suggestions are given below in chronological order. The authors should fully take the following points into consideration and revised their manuscript accordingly, prior to re-submission.

Title: Highlighting the name of nanozyme (core of this manuscript) in the title would be advantageous.

Abstract: 1. Giving citations in the abstract is not a good practice. Remove the citations 1-3 from the abstract (See below for further remark); 2. Write full name for superoxide as superoxide dismutase; 3. Is it right to use abbreviation N-PCNs? or it should be N-PCNSs; 4. Write the name of animal model; 5. The closing lines of an ideal abstract always deal with the possible impact of reported results in future research. Just think about it; 6. The use of term 'our'.. study may be avoided in abstract as well as main text.

Main text:

1. Citation 5 is not relevant to the text. It must be removed from here and adjusted elsewhere, if necessary (Page 1, Line 30)
2. Removed citations 1-3 from the abstract can be placed in line 30 after the text 'possess intrinsic enzyme-like activities'. In addition, proper credit should be given to other recently reported review articles on the field. For instance, *Molecules*, 2015, 20, 14155 and *Molecules*, 2016, 21, 1653.
3. Cited ref. 8 is not a current research. It should be removed (from Page 2, Line 41) and adjusted elsewhere, if necessary.
4. Use abbreviations for iron oxide nanoparticles and N-doped carbon nanomaterials, if possible. For instance, IONPs and N-doped CNMs.
5. The sentence 'thus were designated as....N-PCNs' is reluctant and may be removed (Page 3, Line 77)
6. Figure 1c is a survey scan and not the high resolution spectra for N1s. Do correct it in both text and caption of Figure 1 c-e. (Page 3, Lines 71 and 79) Furthermore, the use of word 'deconvolution' is more appropriate while discussing high-resolution XPS.
7. 398.38 eV and not 398.38 V. Do correct it (Page 3, Line 80).
8. The high resolution C1s spectra is highly desirable for N-PCNs-3. This may be correlated with O1s spectra to ensure the presence of -COOH groups (a requisite for HF_n conjugation). Furthermore, the detailed discussion of both C1s and O1s spectra should be incorporated in the supporting information in order to remove any ambiguity. In this context, the authors should also provide IR data for synthesized nanospheres with or without N-doping.

9. The authors state that 'we synthesized two types of N-PCNs with different N-doping levels' (Page 3, Line 83). However, in the experimental section, N-doping levels are not mentioned but only time (shorter or longer) for carbonization has been found solely responsible for high and low concentration of nitrogen (N)-doping. It should be clarified.
10. Have the authors performed elemental analysis (EA) to determine atomic ratio? Having gone through Table 1 and 2 (SI), I am unable to find the comparative results from EA and XPS. Please provide necessary information with clarity. What do you mean by ID/IG in Table 1?
11. The detailed discussion is highly desired as far as supplementary Fig. 3 or characterization of N-PCNs is considered.
12. The authors state that 'these results clearly indicate....quaternary forms' (Page 3, Lines 87-89). It is confusing. What is the exact morphology of synthesized PCNs? Are they porous or graphitic type? Reconstruction of sentence is necessary for clarity.
13. Supplementary Fig. 4 – Don't you think that incubation time of 30 min. is quite longer? A good nanozyme might show activity in minimal time (max. 10 min.)
14. Figure 2 – revise the caption f and g as it is not clear. Furthermore, I could not see error bars in the figure (last lines in the caption?)
15. In this manuscript, it has been shown that N-Doping is crucial to exhibit four enzyme-like activities (Page 3, Lines 100-101; Page 4, Lines 116-117, and lines 136-137). Do the authors really think that N-doping is the one and only reason for exhibiting enzyme-like activities? If this is so, the carbon nitride dots (having significant amount of nitrogen) might also exhibit four-in-one enzyme-like activity. However, this is not the case. Indeed, it is crucial to explain the factor(s) responsible for the enzyme-like activities of synthesized N-PCNs rather than generalization of N-doping effect.
16. Recheck the line 'as shown in Fig. 2g and 2f'- is this correct? or it should be 2f and 2g.
17. O₂-• - is this a free radical or radical anion? (Page 5, Line 144)
18. Figure 3a – the data for PCNs is also given but x-axis scale deals only with N-PCNs??
19. Figure 3e – kindly provide the name of organic cyclic compound.
20. As mentioned earlier in the abstract, the impact of this research for future work must be included in conclusion in a proper way.

Reviewer #3 (Remarks to the Author):

The work is quite interesting however certain points should be clarified before considering it for publication.

1. According to the author guidelines, references should not be quoted in the abstract.
2. The article is supposed to be divided into separate sections including results and the headings should be used as per the guidelines.
3. The authors are supposed to quote more references in the manuscript, particularly in introduction part. The authors claim to synthesize PCNs and N-PCNs via modified literature method. It is surprising that no reference is quoted for the same (refer to line no. 365-366). Moreover, there is no elaboration of the ambiguities present in the existing literature and how they have overcome the same?
4. The detailed procedure for the synthetic of PCN and N-PCNs as reported by the author is very lengthy and cumbersome. There are several reports for the synthesis of N-doped carbon nanomaterials by low temperature hydrothermal route with very less time.
5. The fate and portal of exit of any nanoparticle given in-vivo with size more than 8nm should be taken under consideration.

6. Although authors have performed enough characterization however the analysis part is lacking in the main manuscript for XRD, Raman spectroscopy and ESR.

7. Authors claim that nitrogen doping and % of oxygen atom is higher for N-PCNs-3 compared to N-PCNs-5. From this data, it is clear that N-PCNs-5 is better reduced form compared to N-PCNs-3, which should reflect in the ID/IG ratio (as referred in supplementary in supplementary table 1) as it is well known fact that graphitic character increases with the removal of oxidizable group. However ID/IG ratio is similar for N-PCNs-5 and N-PCNs-3. Authors should justify for such anomalous behaviour.

8. The authors reported that N-PCNs were capable of collapsing the mitochondrial membrane potential leading to apoptosis which is absent with the undoped carbon nanomaterial. However, there is no data regarding the behaviour of N doped carbon nanomaterials with normal cells. We believe that this experimental data can pave a way for understanding the intricate mechanism behind the toxicity imparted by the N doped nanomaterials.

9. The nanoparticles were administered intratumorally which is not possible for all types of tumor. Can authors kindly elaborate on this aspect of PCNs and N-PCNs?

Reviewer #1 (Remarks to the Author):

The paper of Kelong Fan and colleagues reported the development of a novel nanozyme using nitrogen-doped carbon nanospheres. This nanozyme displays four enzyme-like activities acting as oxidase, peroxidase, catalase and superoxide dismutase and it seems to have a good activity on Reactive Oxygen Species regulation. The nanozyme has been decorated with apoferritin to allow TfR1 mediated uptake and to obtain tumor targeted internalization. Therefore, it has been suggested as anti-cancer agent. The work is nice, well-written and interesting for a broad audience. Nanozyme activities are well-characterized and the idea to target in tumor cells nanozymes using apoferritin is novel.

However, in reviewer's opinion *in vivo* experiments in cell culture and murine model should be refined to make this nanozyme really convincing for clinical applications. Indeed, administration should be performed by systemic route in order to evaluate also nanozyme bioavailability, tumor targeting capability, biodistribution in off-target organs and toxicity, which are not considered in this work but are issues to assess if we want to propose new strategies to treat cancer. Therefore, I suggest to reconsider this paper after major revisions.

Response: We thank the reviewer for the positive comments, and agree with the reviewer's suggestions.

We have now performed *in vivo* antitumor studies using intravenous (*i.v.*) injection of HFn-N-PCNSs-3 to the tumor bearing mice. We also evaluated the biodistribution, biosafety and potential toxicity of HFn-N-PCNSs-3 in tumor-bearing mice and healthy mice. These data have now been added as Figure 4i and 4j in our revised manuscript, as well as Supplementary Figure 17, 18 and 19 in the supplementary information. The related descriptions were added in the revised manuscript (page 9).

Our new results showed that *i.v.* injected HFn-N-PCNSs-3 exhibited significant antitumor activity in both HepG2 and HT-29 tumor models, which confirmed that the HFn-N-PCNSs-3 exhibited excellent tumor targeting capability *in vivo* (Figure 4i-j, Supplementary Figure 17).

To address the biodistribution and toxicity of HFn-N-PCNSs-3 *in vivo*, we performed the pathological analysis of main organs and body weight change analysis of treated mice. Our new data indicated that HFn-N-PCNSs-3 accumulated in liver and spleen (Supplementary Figure 19), and degraded within 15 days (Supplementary Figure 18). No significant changes in body weight were detected in the mice treated with HFn-N-PCNSs-3 (Figure 4j, Supplementary Figure 17d). In addition, no HFn-N-PCNSs were detected in the main organs on Day 30 after *i.v.* injection (Supplementary Figure 19), indicating that HFn-N-PCNSs-3 is an effective and safe antitumor strategy for *in vivo* tumor therapy.

In detail:

1) Figure 3 f: To correctly study colocalization between N-PCNs-3 and Lysosomes a Pearson's coefficient or other correlation studies should be provided.

Response: As suggested, we analyzed the Pearson's co-localization coefficient of N-PCNSs-3 and lysosomes in Figure 3f by Olympus Fluoview Ver.4.1a. The data has been added in the revised manuscript (Page 8).

Moreover LAMP-1 marker is not a Lysosome specific marker since it could be found also in late endosomes (Phillip A. Vanlandingham and Brian P. Ceresa, J Biol Chem. 2009; 284(18): 12110–12124).

Response: To our understanding, late endosomes typically fuse with lysosomes (Hu et al. *Translational Neurodegeneration* (2015) 4:18). In addition, both late endosomes and lysosomes maintain the pH around 4.5-5.5 (acid condition), an important condition that enables N-PCNSs to exhibit their peroxidase and oxidase-like activities'

For improved clarity, we revised the description of the localization of N-PCNSs as late endosome/lysosome in the revised manuscript (page 7).

2) What kind of ferritin is used in this paper? Some works reported the absence of colocalization between H chain ferritin and lysosomes, using Cathepsin D as Lysosome marker. Moreover, they reported that H-chain ferritin as good vector for nuclear delivery (Zhang, L., et al. (2015), *Adv. Healthcare Mater.*, 4: 1305–1310. doi:10.1002/adhm.201500226; Mazzucchelli, S., et al. (2017). *Scientific Reports*, 7(1), 7505. DOI: 10.1038/s41598-017-07617-7; Mazzucchelli S. et al. *Oncotarget*. 2017 Jan 31;8(5):8383-8396. doi: 0.18632/oncotarget.14204.; Bellini M, et al. *J Control Release*. 2014 Dec 28;196:184-96. doi: 10.1016/j.jconrel.2014.10.002. I think you should specify what kind of ferritin you have used here and eventually improve the discussion about lysosomal localization of the Ferritin-decorated nanozyme.

Response: This is an important issue, and we appreciate that the reviewer raised this point. The ferritin used in our study is human H-ferritin (HF_n), whose receptor has been identified as transferrin receptor 1 (TfR1), a typical tumor biomarker.

To address the question whether our HF_n-N-PCNSs localize in the lysosomes, we performed two additional experiments: 1) TEM imaging of the HF_n-N-PCNSs treated cells, and 2) co-localization analysis of HF_n-N-PCNSs and the lysosome marker LysoTracker (Thermo Fisher Scientific) in HepG2 tumor cells. Our new results show that HF_n-N-PCNSs nanoparticles localized in the lysosomes. We did not detect significant levels of nuclear localization of HF_n-N-PCNSs. We added these results into the revised manuscript (Figure 4c and Supplementary Figure 13b).

To our knowledge, the different localization behaviors of HF_n when comparing our study and the reports mentioned by the reviewer might possibly be due to the different pre-treatment of the tumor cells. In our study, we incubated the tumor cells with HF_n-N-PCNSs in the normal culture conditions. In addition, others' and our own

previous work have already showed that HFn nanoparticles localized in the lysosomal compartment in many types of tumor cells (*PNAS*, 2010, 107(8): 3505-3510; *PNAS*, 2014, 111(41):14900-14905). However, for the nuclear delivery of HFn, all tumor cells reported in those works (reviewer mentioned) were pre-treated by different conditions. For example, in the work of Zhang and colleagues, tumor cells were pre-treated using desferrioxamine (DFO), rendering these tumor cells in iron-deficiency condition; In the work of Mazzucchelli *et al.*, treatment with the drug olaparib triggered the nuclear translocation of endogenous HFn, thus the nuclear localization of HFn-olaparib is mediated by the released olaparib from the degraded HFn nanocarrier in lysosome (*Scientific Reports*, 7(1), 7505). Similarly, in the studies of Mazzucchelli *et al.* and Bellini *et al.* HFn nanocarriers were promptly translocated into the nucleus following the DNA damage caused by the partial released doxorubicin from HFn nanocarriers in cytoplasm (*J Control Release*. 2014, 28;196; *Oncotarget*. 2017, 8(5):8383-8396).

In the new version of manuscript, we added the discussion about taking advantage of the nanocarrier property of HFn further improving the anti-tumor ability using the developed nanozyme strategy for tumor catalytic therapy (page 9).

3) Figure 4 c: It is not possible study colocalization between a fluorescence signal acquired in confocal microscopy and the nanozyme signal acquired in bright field since the last one has not an adequate spatial definition along the z-axis.

Response: As suggested, we performed the TEM analysis of localization of HFn-N-PCNSs in tumor cells, and revised the Figure 4c.

4) Figure 4e-f-g-h: The displayed results are very interesting but have been obtained with intratumoral injection of the Nanozymes. I think this is the main bias of the work. In this way you demonstrate the efficacy on the tumor but you cannot demonstrate “safety” for healthy cells and tumor targeting capability in vivo. It should be very nice to see the same results obtained after an i.v. administration in order to evaluate also bioavailability and biodistribution. Moreover, the in vivo efficacy should be confirmed using other tumor models.

Response: Please see our response to the major comments of Reviewer #1. We further evaluated the antitumor activity of HFn-N-PCNSs by employing human colon cancer HT-29 xenograft tumor model. We have now added these results as Supplementary Figure 17 in the revised supplementary information. Related descriptions were added in the revised manuscript (page 9).

5) Supp. Figure 10. The demonstration of tumor targeted internalization by bright field microscope images is not convincing. You could perform an uptake study by flow cytometry using fluorescent labelled ferritin-decorated nanozymes or a confocal microscopy analysis.

Response: We thank the reviewer for this suggestion. We performed the flow cytometry analysis for the internalized HFn-N-PCNSs and N-PCNSs in tumor cells, and the results were shown in Figure 4d.

Minor remark:

1) Figure 3 c: What is CCCP? I suppose it is the positive control. I think it should be better increase figure readability describing the meaning of green and red spots.

Response: As suggested by the reviewer, we have now explained CCCP as well as the meaning of green and red colors in the methods section of the revised manuscript (page 17).

2) Line 156-157: “indicating that PCNs trigger an apoptotic cascade inside the targeted cells only after N-doping. The use of the term “only” is not appropriate: there are green spots also in PCNs sample.

Response: As suggested, we deleted ‘only’ in the revised manuscript (page 6).

3) Line 227: “...healthy organs remained undamaged...” Of course, you have not performed a systemic administration! To assess systemic toxicity you should perform a systemic administration.

Response: We thank the reviewer for pointing out this flaw in our argument. We have repeated the animal experiment using *i.v.* administration. Our results showed that there was no systemic toxicity for N-PCNSs. Furthermore, N-PCNSs was degradable for long-term incubation in aqueous solution, thus reducing the safety risk. Please also see the response to the major comments of Reviewer #1.

4) Methods section: western blot analysis reported in Supp. Fig. 11 is not described.

Response: We apologize for this omission, we have now added the description of western blot analysis in the revised Methods section (page 19).

5) Supp. Figure 11. A viability assay on healthy TfR1 expressing cell line should be performed to clarify what happens in healthy cells with high TfR1 expression such as endothelial cells.

Response: As suggested by the reviewer, we have now performed a biosafety analysis *in vivo*, whereby we evaluated the potential side-effects in the main organ of healthy mice. No significant pathological changes in main organs of healthy mice treated with HFn-N-PCNSs-3 were detected.

The results were added as Figure 4j, Supplementary Figure 17d and Supplementary Figure 19 in the revised manuscript and supplementary information.

Reviewer #2 (Remarks to the Author):

This manuscript authored by Yan, Gao and co-workers describes the potential of N-doped carbon nanospheres (N-PCNs), in particular, N-PCNs-3 as four-in-one nanozyme for in vivo tumor catalytic therapy. Given the interesting as well as surprising features of N-PCNs (in terms of exhibiting four different enzyme-like activities), indubitably, this is an excellent research work in the fields of bioanalytical and biomedical sciences as well as carbon-based nanomaterials (NMs).

However, in view of several open questions in conjunction with inadequate characterization of reported material(s), insufficient and/or inappropriate citation(s), and contemptible text (at many instances), I personally feel that the manuscript is not suitable for publication in its present form. Consequently, I recommend its publication in Nature Communications subjected to major revisions. For authors' reference, some of my remarks/suggestions are given below in chronological order. The authors should fully take the following points into consideration and revised their manuscript accordingly, prior to re-submission.

Title: Highlighting the name of nanozyme (core of this manuscript) in the title would be advantageous.

Response: We thank the reviewer for the positive comments. As suggested, we have now changed our title to 'In vivo Guiding Nitrogen-doped Carbon Nanozyme for Tumor Catalytic Therapy'.

Abstract: 1. Giving citations in the abstract is not a good practice. Remove the citations 1-3 from the abstract (See below for further remark);

Response: We have removed the references 1-3 from the abstract to the introduction section, and revised them as references 2,3,6 in the revised manuscript (page 1).

2. Write full name for superoxide as superoxide dismutase;

Response: We apologize for this omission, we have now revised this term.

3. Is it right to use abbreviation N-PCNs? or it should be N-PCNSs;

Response: We thank the reviewer for this suggestion, and have now changed the abbreviation throughout the revised manuscript.

4. Write the name of animal model;

Response: As suggested, we have now included the complete name of our animal model (human tumor xenograft mice model) in the abstract of the revised manuscript.

5. The closing lines of an ideal abstract always deal with the possible impact of reported results in future research. Just think about it;

Response: As suggested, we have now added a sentence on the potential impact in future research in the revised abstract.

6. The use of term 'our'.. study may be avoided in abstract as well as main text.

Response: As suggested, we deleted all the 'our' descriptions in the revised manuscript.

Main text:

1. Citation 5 is not relevant to the text. It must be removed from here and adjusted elsewhere, if necessary (Page 1, Line 30)

Response: As suggested, we revised the citation 5 as reference 8, and moved it to the second paragraph from bottom on page 1.

2. Removed citations 1-3 from the abstract can be placed in line 30 after the text 'possess intrinsic enzyme-like activities'. In addition, proper credit should be given to other recently reported review articles on the field. For instance, *Molecules*, 2015, 20, 14155 and *Molecules*, 2016, 21, 1653.

Response: We apologize for these omissions. We have now removed the references 1-3 from the abstract to the introduction section, and revised them as references 2,3,6 in the revised manuscript in Page 1. Also, we have added two new references as references 4 and 5 in the revised page 1.

3. Cited ref. 8 is not a current research. It should be removed (from Page 2, Line 41) and adjusted elsewhere, if necessary.

Response: As suggested, we have deleted this citation.

4. Use abbreviations for iron oxide nanoparticles and N-doped carbon nanomaterials, if possible. For instance, IONPs and N-doped CNMs.

Response: We thank the reviewer for this suggestion, and hope that the amount of abbreviations used in our paper remains within the limits of the journal guidelines. We have now introduced the abbreviations IONPs and N-CNMs to represent iron oxide

nanoparticles and N-doped carbon nanomaterials, respectively.

5. The sentence 'thus were designated as...N-PCNs' is reluctant and may be removed (Page 3, Line 77)

Response: As suggested, we deleted this sentence in the revised page 3.

6. Figure 1c is a survey scan and not the high resolution spectra for N1s. Do correct it in both text and caption of Figure 1 c-e. (Page 3, Lines 71 and 79) Furthermore, the use of word 'deconvolution' is more appropriate while discussing high-resolution XPS.

Response: We corrected the expression of XPS profiles in both text and caption of Figure 1c –e.

7. 398.38 eV and not 398.38 V. Do correct it (Page 3, Line 80).

Response: We have corrected this error in the revised manuscript.

8. The high resolution C1s spectra is highly desirable for N-PCNs-3. This may be correlated with O1s spectra to ensure the presence of –COOH groups (a requisite for HFn conjugation). Furthermore, the detailed discussion of both C1s and O1s spectra should be incorporated in the supporting information in order to remove any ambiguity. In this context, the authors should also provide IR data for synthesized nanospheres with or without N-doping.

Response: We thank the reviewer's suggestion. We have now added a detailed discussion of the C 1s and O 1s spectra in the supplementary information, with NR data included in Supplementary Figure 4.

9. The authors state that 'we synthesized two types of N-PCNs with different N-doping levels' (Page 3, Line 83). However, in the experimental section, N-doping levels are not mentioned but only time (shorter or longer) for carbonization has been found solely responsible for high and low concentration of nitrogen (N)-doping. It should be clarified.

Response: As suggested, we clarified this in the Methods section in page 14 of revised manuscript.

10. Have the authors performed elemental analysis (EA) to determine atomic ratio? Having gone through Table 1 and 2 (SI), I am unable to find the comparative results from EA and XPS. Please provide necessary information with clarity. What do you mean by ID/IG in Table 1?

Response: We apologize for the unclear description of our data. We calculated the carbon, oxygen and nitrogen contents using the XPS data. We have now corrected the title of Supplementary Table 1. The I_D/I_G ratios were obtained from Raman spectra, and we added corresponding discussions in Page 3-4 of the revised supplementary information.

11. The detailed discussion is highly desired as far as supplementary Fig. 3 or characterization of N-PCNs is considered.

Response: As suggested, the detailed discussion of the characterization of N-PCNSs have been added in page 3-4 of the revised supplementary information.

12. The authors state that 'these results clearly indicate....quaternary forms' (Page 3, Lines 87-89). It is confusing. What is the exact morphology of synthesized PCNs? Are they porous or graphitic type? Reconstruction of sentence is necessary for clarity.

Response: As suggested, this section was re-edited in page 3 of the revised manuscript.

13. Supplementary Fig. 4 – Don't you think that incubation time of 30 min. is quite longer? A good nanozyme might show activity in minimal time (max. 10 min.)

Response: We thank the reviewer for this question. The reaction time usually depends on the nanozyme amount as well as on the substrate concentration. For oxidase-like activity, the oxygen is the substrate, but its concentration in the solution is low due to the dissolved oxygen in water. Thus, oxygen concentration is a limiting factor for the reaction rate. We found that the colorimetric reaction proceeded at an accelerated rate in the oxygen atmosphere (please see the revised Supplementary Figure 5d). Therefore, increased O_2 levels result in higher reaction rates.

14. Figure 2 – revise the caption f and g as it is not clear. Furthermore, I could not see error bars in the figure (last lines in the caption?)

Response: As suggested, the caption f and g have been re-edited for clarity. All data were performed in triplicate and had error bar. The last line in each caption represents the data from the group with very low or no enzyme-like activity. Therefore, the value of the data is very small and the corresponding error bar in the figure is inconspicuous. We have now revised the size of the symbols in Figure 2, hopefully to good effect.

15. In this manuscript, it has been shown that N-Doping is crucial to exhibit four enzyme-like activities (Page 3, Lines 100-101; Page 4, Lines 116-117, and lines 136-137). Do the authors really think that N-doping is the one and only reason for exhibiting enzyme-like activities? If this is so, the carbon nitride dots (having significant amount of nitrogen) might also exhibit four-in-one enzyme-like activity.

However, this is not the case. Indeed, it is crucial to explain the factor(s) responsible for the enzyme-like activities of synthesized N-PCNs rather than generalization of N-doping effect.

Response: We agree with the reviewer that understanding the mechanisms for these enzyme-like activities of N-PCNSs is of tremendous interest. To check if N-doping alone is sufficient to initiate catalysis, we chose a molecule, flavin adenine dinucleotide (FAD) which has similar typical pyridinic N and quaternary N as those in N-PCNSs-3, to test the enzyme-like activities. The result showed that FAD did not perform any activities neither for oxidase, peroxidase nor catalase (Supplementary Figure 9), indicating that: N alone is not sufficient to promote catalysis.

On the basis of our experimental results, we propose that both N-doping as well as the graphitic structure of N-PCNSs are necessary factors to induce enzyme-like activities. To confirm the importance of graphitization, we analyzed the activity of the precursor (before carbonization) of N-PCNSs-3 and found that the activity in the precursor was negligible (Supplementary Figure 8), indicating that the graphitization step is necessary to promote catalysis.

Carbon nitride (C_3N_4) dots possess similar N-doping as those in our N-PCNSs. However, C_3N_4 are often prepared at 450-580°C (Chen J, et al., *ACS Nano*. 2017 2017, 11 (12): 12650–12657; Nasir M, et al. *Microchim Acta* (2017) 184: 323-342.), and thus the degree of graphitization ($I_D/I_G > 1$) is lower. We assume that because of this low degree of graphitization, carbon nitride may not show similar enzyme activities as those of our N-PCNSs. The same difference may be found for N-doping carbon dots, which have many N-dopants but still show limited activity due to insufficient graphitization at low carbonization temperatures (~200°C) (Yang W, et al. *Talanta*, 2017, 164:1-6). In our experiment, we prepared N-PCNSs for carbonization at 800°C which resulted in a high degree of graphitization.

Thus, we believe that the high enzyme-like activities observed for our N-PCNSs may be a result of the high level of graphitization, in addition to N-doping.

Currently, we are systematically comparing the activities between multiple carbon materials with different N-doping conditions, including the types of N in the carbon nanostructure and graphitization level, to determine in more detail the precise role of doped N and graphitic structure in the enzyme-like activities.

16. Recheck the line 'as shown in Fig. 2g and 2f'- is this correct? or it should be 2f and 2g.

Response: As suggested, it is corrected in the revised manuscript.

17. O_2^- - is this a free radical or radical anion? (Page 5, Line 144)

Response: O_2^- represents the superoxide radical, one of the reactive oxygen species. Thus, it is a free radical in the superoxide anion.

18. Figure 3a – the data for PCNs is also given but x-axis scale deals only with N-PCNs??

Response: As suggested, the X-axis of Figure 3a was corrected.

19. Figure 3e – kindly provide the name of organic cyclic compound.

Response: We appreciate this suggestion. To make this issue more clearly, we added the explanation of the ‘organic cyclic compound’ in Figure 3e as ‘cellular response to organic cyclic compound’ in the figure legend in revised manuscript in Page 8.

As the matter of fact, the RNA seq analysis only provided the information of the differential expression profiling of the cells treated with different compounds. In this study, the number of mRNA changes in the biological process regarding to the cellular response to organic cyclic compound affected by N-PCNs is 66, which is shown in Figure 3e.

20. As mentioned earlier in the abstract, the impact of this research for future work must be included in conclusion in a proper way.

Response: As suggested, we revised the last sentence of the conclusion section in the revised manuscript.

Reviewer #3 (Remarks to the Author):

The work is quite interesting however certain points should be clarified before considering it for publication.

1. According to the author guidelines, references should not be quoted in the abstract.

Response: We thank the reviewer for the encouraging comments. As suggested, we have removed the references 1-3 from the abstract to the introduction section in the revised manuscript.

2. The article is supposed to be divided into separate sections including results and the headings should be used as per the guidelines.

Response: We apologize for the formatting issues. We have now divided the results section into three parts, and added the headings for each part in the revised manuscript.

3. The authors are supposed to quote more references in the manuscript, particularly in introduction part. The authors claim to synthesize PCNs and N-PCNs via modified literature method. It is surprising that no reference is quoted for the same (refer to line no. 365-366). Moreover, there is no elaboration of the ambiguities present in the existing literature and how they have overcome the same?

Response: We thank the reviewer for pointing out the lack of references. We have now added all relevant references suggested by the reviewer, in the Results section (page 3) and Methods section (page 14).

We synthesized the PCNSs and N-PCNSs according to the work reported by Prof. Dongyuan Zhao (Ref. 21, A Low-Concentration Hydrothermal Synthesis of Biocompatible Ordered Mesoporous Carbon Nanospheres with Tunable and Uniform Size. *Angew Chem Int Edit* 2010, 49: 7987-7991). In Zhao's paper, the triblock copolymer Pluronic F127 was used as a template, and a phenolic resol was used as a carbon source. The carbonization was carried out for 3 h at 700°C in an oxygen-free N₂ atmosphere. Our work differs from the literature primarily in adding melamine to achieve nitrogen doping. In addition, to increase graphitization levels, we carbonized the N-PCNSs at 800°C. The main reason for this change is that graphitization of the carbon materials is incomplete at lower temperatures, resulting poor enzyme-like activities.

We have now revised the paragraph concerning material synthesis (page 3 and 14).

4. The detailed procedure for the synthetic of PCN and N-PCNs as reported by the author is very lengthy and cumbersome. There are several reports for the synthesis of N-doped carbon nanomaterials by low temperature hydrothermal route with very less time.

Response: While our system is lengthier than comparable methods, one major advantage of our synthesis procedure is that it produces highly graphitized PCNSs. N heteroatom-doping and peripheral graphitic structure are both important in the enzyme-like activities of carbon-based materials (discussed in our revised manuscript in page 5-6).

In contrast, the carbon materials obtained through low temperature hydrothermal route used in other reports, such as N-doped carbon dots but the graphitization is not sufficient to ensure products with high enzyme-like activity.

Considering the practical applications in the future, however, we agree with the reviewer that it will be important to optimize the preparation procedure and to make it less cumbersome. Our paper presents a proof-of-principle study, and as such we hope it will inspire other laboratories to join the search for improved methodologies.

5. The fate and portal of exit of any nanoparticle given in-vivo with size more than 8nm should be taken under consideration.

Response: We thank the reviewer for this question. To investigate the fate and portal of exit of N-PCNSs, we performed the biodegradation and biosafety analysis of N-PCNSs both *in vitro* and *in vivo*. The results were added as Figure 4j, Supplementary Figure 17d, Supplementary Figure 18 and 19 in the revised manuscript and supplementary information. These new data indicated that HFn-N-PCNSs is biodegradable *in vivo*, and exhibited no toxicity to healthy organs.

6. Although authors have performed enough characterization however the analysis part is lacking in the main manuscript for XRD, Raman spectroscopy and ESR.

Response: According to the suggestion, we added descriptions of XRD, Raman spectroscopy and ESR in the support information.

7. Authors claim that nitrogen doping and % of oxygen atom is higher for N-PCNs-3 compared to N-PCNs-5. From this data, it is clear that N-PCNs-5 is better reduced form compared to N-PCNs-3, which should reflect in the ID/IG ratio (as referred in supplementary in supplementary table 1) as it is well known fact that graphitic character increases with the removal of oxidizable group. However I_D/I_G ratio is similar for N-PCNs-5 and N-PCNs-3. Authors should justify for such anomalous behaviour.

Response: Thanks for pointing out the critical issue. We agree that the increased carbonization time would increase the graphitic character of carbon materials. We found that the I_D/I_G for N-PCNSs-5 was 0.925, which was lower than that for N-PCNSs-3 (0.926), but the difference is negligible.

We assume that the high carbonization temperature primarily resulted in high level of graphitization. Both the graphitization in N-PCNSs-3 and N-PCNSs-5 have reached the plateau quickly at 800°C. Longer time carbonization under this temperature may remove more nitrogen and oxygen from N-PCNSs, but it cannot change the graphitization level dramatically. Therefore, although the N-doping and oxygen ratio in N-PCNSs-5 is lower than those in N-PCNSs-3, their graphitization is already in the highest level, resulting in the similar ID/IG ratio.

Because nitrogen is also removed during the carbonization reaction, it is essential to improve the carbonization conditions further to ensure sufficient N-doping and graphitization. We will keep investigating the optimal conditions for future study.

8. The authors reported that N-PCNs were capable of collapsing the mitochondrial membrane potential leading to apoptosis which is absent with the undoped carbon nanomaterial. However, there is no data regarding the behaviour of N doped carbon nanomaterials with normal cells. We believe that this experimental data can pave a way for understanding the intricate mechanism behind the toxicity imparted by the N doped nanomaterials.

Response: We thank the reviewer for this comment, and we completely agree that it is

paramount to investigate the molecular details.

As shown in Supplementary Figure 14, N-PCNSs induced significant cell death for both tumor cells and normal cells, because N-PCNSs themselves cannot distinguish tumor cells from normal cells. To allow our N-PCNSs to specifically target tumor cells, we chose HFn as navigator to guide the targeting of N-PCNSs *in vitro* and *in vivo*. As shown in Supplementary Figure 14, after modification with HFn, HFn-N-PCNSs exhibited no toxicity to normal cells (hASMCs) characterized by low TfR1 surface expression. In comparison, accumulation of HFn-N-PCNSs improved greatly in different types of tumor cells (Figure 4 and Supplementary Figure 17).

For clarity, we added the subtitle of ‘Ferritin coordinating N-PCNSs for *in vivo* tumor catalytic therapy’ on Page 8 of the revised manuscript.

9. The nanoparticles were administered intratumorally which is not possible for all types of tumor. Can authors kindly elaborate on this aspect of PCNs and N-PCNs?

Response: We thank the reviewer for pointing out this potential issue. As we did not test this point for our original study, we evaluated the antitumor activity of HFn-N-PCNSs *in vivo* via intravenously (*i.v.*) injection for two human tumor xenograft tumor models. We found that that *i.v.* injected HFn-N-PCNSs-3 exhibit significant antitumor activity in both HepG2 and HT-29 tumor model. Thus, we demonstrate that HFn-N-PCNSs effectively treat the tumors *in vivo* via both intratumoral administration and intravenous administration.

We have now added these results to our revised manuscript (Figure 4i-j and Supplementary Figure 17).

Reviewers' comments:

Reviewer #1 (Remarks to the Author):

In my opinion, the revised manuscript answers all the issues raised from reviewers. The manuscript is now suitable for publication in Nature Communication.

Reviewer #2 (Remarks to the Author):

Recommendation or Remarks: Minor Revisions

Comments to Authors

Having gone through the revised version of the manuscript entitled, 'In vivo Guiding Nitrogen-doped Carbon Nanozyme for Tumor Catalytic Therapy', I have realized that the authors have given a good response against my queries/suggestions and most of the issues are satisfactorily addressed. Nevertheless, I feel that the revised version still has some faintness, and my remarks for the same are appended below.

1. Supplementary Fig. 4: The authors state that the 'FT-IR analysis showed...carboxyl group'. But C-H (1037 cm^{-1}) is not an oxygen-containing group. Please correct this error. Most importantly, if I correlate the XPS data, I could not see any IR vibration for -COOH groups (XPS: O-C=O at 288.0 eV) in Supplementary Fig. 4. If the synthesized nanodots are of graphitic type (as emphasized by the authors at many instances in the revised version), the IR spectra might show an intense or at least moderate band around 1710-1730 cm^{-1} for -COOH functions. Further, if one compares the IR spectra of PCNSs and N-PCNSs, a severe attenuation in the C=O bands is quite observable. This indicates that the carbonization at high temperature (800°C in the present case) can easily remove the oxygen-containing functional groups as can be seen with two dimensional graphite oxide and/or graphene oxide. Can authors provide a convincing explanation for these issues?

2. Kindly consider these two statements of yours in the revised version- (a) 'that nitrogen...porous carbon framework of PCNSs' as also characterized by TEM and SEM [Page 4, lines 96 & 97] and (b) 'we conclude that both N-doping and peripheral graphitic structure are important...'. Now, don't you think that these two statements are contradictory? Furthermore, can you really use the phrase, "peripheral graphitic structure"? Indeed, the inner part of nanodots is primarily composed of sp^2 -hybridized carbon atoms (graphitic feature) while sp^3 -hybridized carbon atoms are present in the outer part as also evidenced by NMR analysis (Angew. Chem. Int. Ed., 2007, 46, 6473). What do you think?

3. In response to query no. 15 (as per the submitted response letter), the authors state that 'thus, we believe....in addition to N-doping' (Page 9, Paragraph 4). I would like to mention that the nitrogenous reagents (for ex. Melamine in the present case) can serve as both N-source (N-doping) and surface passivation agent. Under such complexity, how would you differentiate between the former and later?

4. Please correct the word 'N-PCNs' as 'N-PCNSs' in the caption of supplementary Figure 8 (Page 8, Line 149)

5. I could not see any change as far as query no. 18 (as per the submitted response letter) is considered. The X-axis of Figure 3a should be corrected as PCNSs/N-PCNSs ($\mu\text{g/mL}$).

6. Again, no revision has been made against query no. 20 (last one). It is same as before. Please

highlight the impact of this research in a more scientific way, if possible.

Reviewer #3 (Remarks to the Author):

I appreciate the authors to revise the manuscript in the line of the comments. The manuscript can be considered for publication after incorporation of plausible answers of the following comments.

1. The precursor material is porous in nature as reported in Yin Fang et al. *Angew. Chem. Int. Ed.* (2010) 49, 7987 –7991. Authors have added melamine and undergone carbonisation at high temperature for the development of N-PCNSs. Such changes can alter porosity of the N-PCNSs. So BET Isotherm should be performed to get further insight about surface area and porous structure.

2. “Longer time carbonization under this temperature may remove more nitrogen and oxygen from N-PCNSs, but it cannot change the graphitization level dramatically”— Authors should provide few references to support their hypothesis.

Reviewer #2 (Remarks to the Author):

Having gone through the revised version of the manuscript entitled, 'In vivo Guiding Nitrogen doped Carbon Nanozyme for Tumor Catalytic Therapy', I have realized that the authors have given a good response against my queries/suggestions and most of the issues are satisfactorily addressed. Nevertheless, I feel that the revised version still has some faintness, and my remarks for the same are appended below.

1. Supplementary Fig. 4: The authors state that the 'FT-IR analysis showed...carboxyl group'. But C-H (1037 cm^{-1}) is not an oxygen-containing group. Please correct this error.

Response: As suggested, we have removed C-H (1037 cm^{-1}) in the revised Supplementary Fig. 4.

Most importantly, if I correlate the XPS data, I could not see any IR vibration for –COOH groups (XPS: O-C=O at 288.0 eV) in Supplementary Fig. 4. If the synthesized nanodots are of graphitic type (as emphasized by the authors at many instances in the revised version), the IR spectra might show an intense or at least moderate band around 1710-1730 cm^{-1} for –COOH functions. Further, if one compares the IR spectra of PCNSs and N-PCNSs, a severe attenuation in the C=O bands is quite observable. This indicates that the carbonization at high temperature (800°C in the present case) can easily remove the oxygen-containing functional groups as can be seen with two dimensional graphite oxide and/or graphene oxide. Can authors provide a convincing explanation for these issues?

Response: We thank the reviewer for this question. We carefully re-analyzed the FTIR spectra of PCNSs and N-PCNSs. As shown in revised Supplementary Fig. 4, the C=O bands located at 1712 cm^{-1} were observed in N-PCNSs. Moreover, the band intensity

of C=O decreased with the prolong carbonation time in N-PCNSs-5 compared to N-PCNSs-3. The low signal of carboxyl group may be related to the total oxygen content in PCNSs. All oxygen contents dramatically decreased during carbonization under 800°C, of which were determined as 6.34, 4.58 and 4.06 atom% for PCNSs, N-PCNSs-3 and N-PCNSs-5, respectively. Actually, the majority of oxygen may have been removed below 400°C by pyrolysis (*Appl. Catal. B-Environ.* 2014, 147, 369-376). Therefore, the remaining oxygen content in PCNSs becomes low after carbonization.

Importantly, the removal of oxygen may facilitate nitrogen doping as it may generate active sites for nitrogen incorporation into carbon framework (*ACS Nano* 2011, 5, 4350-4358). There was 6.34 atom% oxygen in PCNSs, while N-PCNSs-3 contained 4.58 atom% of oxygen and 3.37 atom% of nitrogen. These data indicate that nitrogen doping may lead to more oxygen removal, which is in agreement with the phenomena for nitrogen-doped graphene oxide in Xia's and Dai's previous reports (*ACS Nano* 2011, 5, 4350-4358; *J. Am. Chem. Soc.* 2009, 31, 15939-15944). In addition, 4.06 atom% of oxygen was determined in N-PCNSs-5, while that in N-PCNSs-3 was 4.58 atom%, indicating that longer carbonization time removed more oxygen. Taken together, carbonization under high temperature (800°C) resulted in easy removal of oxygen. Also, nitrogen doping increased the removal of oxygen in N-PCNSs. The weak signal of carboxyl group in the characterization of N-PCNSs may be ascribed to such loss of total oxygen content.

2. Kindly consider these two statements of yours in the revised version- (a) 'that nitrogen...porous carbon framework of PCNSs' as also characterized by TEM and SEM [Page 4, lines 96 & 97] and (b) 'we conclude that both N-doping and peripheral graphitic structure are important....' Now, don't you think that these two statements are contradictory? Furthermore, can you really use the phrase, "peripheral graphitic

structure”? Indeed, the inner part of nanodots is primarily composed of sp²-hybridized carbon atoms (graphitic feature) while sp³-hybridized carbon atoms are present in the outer part as also evidenced by NMR analysis (Angew. Chem. Int. Ed., 2007, 46, 6473). What do you think?

Response: We really appreciate that the reviewer pointed out this issue. The purpose we used the word “peripheral” in the main text is to indicate that the catalysis occurs on the surface of N-PCNSs. According to the reports for other nanozymes, nanocatalysts and natural enzymes, most active centers locate on the surface layer which provides an interface to allow substrate adsorption and product dissociation quickly. Our original intention is to use “peripheral” to refer the “surface”. We really thank the reviewer pointing out the knowledge for peripheral graphitic carbon atoms. To make it clear, we have deleted the word “peripheral” in the main text. Please see it in page 6 in the revised manuscript.

3. In response to query no. 15 (as per the submitted response letter), the authors state that ‘thus, we believe....in addition to N-doping’ (Page 9, Paragraph 4). I would like to mention that the nitrogenous reagents (for ex. Melamine in the present case) can serve as both N-source (N-doping) and surface passivation agent. Under such complexity, how would you differentiate between the former and later?

Response: We thank the reviewer for this question. Melamine is a compound containing three amine groups with a 1, 3, 5-triazine skeleton. It contains 67% nitrogen by mass, and thus it is a good N-source in the synthesis of nanomaterials (Chem. Mater., 2010, 22, 428-434; J. Phys. Chem. C, 2014, 118, 2507-2517; ChemSusChem, 2013, 6, 807-812). It is a precursor to fabricate carbon nitride (C₃N₄) nanodots in which C/N atom ratio is often up to the theoretical value 0.75 (ACS Appl. Mater. Interfaces, 2014, 6, 1258-1265). In our N-PCNSs preparation, we used melamine as the N-source to dope into carbon framework under 800°C which is much

higher than that used for carbon nitride. The N content measured by XPS is 3.37% in N-PCNSs-3 (please see Supplementary Table 1) with pyridinic nitrogen (N-6), pyrrolic nitrogen (N-5), quaternary nitrogen (N-Q) and pyridine oxide or the oxidized nitrogen (N-O_x) groups which are present for nitrogen-doping with melamine in graphene materials (*Appl. Catal. B-Environ.*, 2014, 147, 369-376). If melamine acts as surface passivation agent on carbon nanospheres, it may form N-rich surface after carbonization. For instance, carbon dots with melamine for surface passivation are doped with nitrogen up to 54.98% (*Sensor. Actuat. B-Chem.*, 2018, 255: 1130-1138). The low N content in our N-PCNSs indicates that melamine is the N source for N-doping.

We also determined the oxidase-like and peroxidase-like activities of the complex of PCNSs (without N-doping) with free melamine on the surface (Figures only for reviewer). As expected, no enzyme activity of this complex was detected, indicating that melamine adsorbing on the surface of PCNSs does not have catalytic activity. These results demonstrated that nitrogen needs to be doped into carbon framework to render PCNSs possessing the enzyme-like activity. Therefore, based on the above results and literatures, we think that melamine acts as the nitrogen source for N-doping to carbon framework in this study. As the N-doping and types of nitrogen in carbon framework is important for the catalytic activity of carbon nanozymes, in future research we will keep studying the doping condition and identification to optimize the catalytic performance.

Figures only for reviewer. Activity study in melamine/PCNSs system. (a) The POD-like activity of in the reaction containing 0.416 mM TMB and 0.5292M H₂O₂. 1: 25 µg/mL PCNSs; 2: 25 µg/mL PCNSs + 5 µg/mL melamine; 3: 25 µg/mL PCNSs + 10 µg/mL melamine; 4: 25 µg/mL PCNSs + 25 µg/mL melamine. (b) The OXD-like activity of in the reaction containing 0.416 mM TMB. 1: 25 µg/mL PCNSs; 2: 25 µg/mL PCNSs + 5 µg/mL melamine; 3: 25 µg/mL PCNSs + 10 µg/mL melamine; 4: 25 µg/mL PCNSs + 25 µg/mL melamine.

4. Please correct the word 'N-PCNs' as 'N-PCNSs' in the caption of supplementary Figure 8 (Page 8, Line 149)

Response: As suggested, we have corrected it in the revised manuscript.

5. I could not see any change as far as query no. 18 (as per the submitted response letter) is considered. The X-axis of Figure 3a should be corrected as PCNSs/N-PCNSs (µg/mL).

Response: As suggested, we have corrected it in the revised manuscript.

6. Again, no revision has been made against query no. 20 (last one). It is same as before. Please highlight the impact of this research in a more scientific way, if possible.

Response: We apologize for this omission. We have revised the summary section with discussion as reviewer suggested. We highlighted the impact of our work in developing artificial enzymes for *in vivo* application, the approach to guiding a nanozyme to a living cell and extending the applications of carbon nanocatalysts in biomedicine.

Reviewer #3 (Remarks to the Author):

I appreciate the authors to revise the manuscript in the line of the comments. The manuscript can be considered for publication after incorporation of plausible answers of the following comments.

1. The precursor material is porous in nature as reported in Yin Fang et al. Angew. Chem. Int. Ed. (2010) 49, 7987 –7991. Authors have added melamine and undergone carbonisation at high temperature for the development of N-PCNSs. Such changes can alter porosity of the N-PCNSs. So BET Isotherm should be performed to get further insight about surface area and porous structure.

Response: We thank the reviewer's encouraging comments and suggestions. We performed BET Isotherm assay to get further insight about surface area and porous structure. As shown in Supplementary Fig. 5, the surface area of PCNSs, N-PCNSs-3 and N-PCNSs-5 was 583.8 , 542.1 and 614.8 m²/g, respectively. The pore sizes of PCNSs, N-PCNSs-3 and N-PCNSs-5 were determined as 2.2, 3.0 and 3.1 nm, respectively. Therefore, the addition of melamine did not cause obvious difference in surface area and porous structure of PCNSs.

2. “Longer time carbonization under this temperature may remove more nitrogen and oxygen from N-PCNSs, but it cannot change the graphitization level dramatically”—
Authors should provide few references to support their hypothesis.

Response: We thank the reviewer’s suggestion. The hypothesis is proposed based on our experimental data and other literatures. In our experiment, we found that the I_D/I_G for PCNSs, N-PCNSs-3 and N-PCNSs-5 were 0.934, 0.926, and 0.925 (Supplementary Table 1), respectively, which indicates that both PCNSs and N-PCNSs achieved same high level of graphitization through the carbonization process under 800°C. In comparison, the oxygen content was reduced in N-PCNSs (4.58 atom% for N-PCNSs-3 and 4.06 atom% for N-PCNSs-5) compared to PCNSs (6.34 atom%). This result is consistent with the phenomena that high temperature makes the oxygen easy removal during carbonization process (*Appl. Catal. B-Environ.* 2014, 147, 369-376; *ACS Nano* 2011, 5, 4350-4358).

Moreover, the nitrogen content of N-PCNSs also decreased under longer time incubation at 800°C. Our data showed that the nitrogen in N-PCNSs-3 was 3.37 atom% and 2.85 atom% in N-PCNSs-5, indicating nitrogen removal occurred over time under high temperature. Similar phenomena were observed in the study for nitrogen doping in graphene (*Nanoscale*, 2016, 8, 2795-2803; *Energy Environ. Sci.*, 2014, 7, 1212-1249). Therefore, we speculate the graphitic structure is more stable than heteroatomic nitrogen and oxygen under pyrolysis. We will keep studying the related factor and mechanism with our N-PCNSs in the future.

REVIEWERS' COMMENTS:

Reviewer #2 (Remarks to the Author):

The authors have addressed all issues (except remark 1) at their level best. Though I still have some reservations w.r.t. remark 1 yet I do not think that it should preclude the publication of this manuscript, and consequently, I strongly recommend the publication of this manuscript in Nature communications.

Reviewer #3 (Remarks to the Author):

The paper can be accepted in this form.

REVIEWERS' COMMENTS:

—
Reviewer #2 (Remarks to the Author):

—
The authors have addressed all issues (except remark 1) at their level best. Though I still have some reservations w.r.t. remark 1 yet I do not think that it should preclude the publication of this manuscript, and consequently, I strongly recommend the publication of this manuscript in Nature communications.

Response: We thank the reviewer for the encouraging and positive comments.

Reviewer #3 (Remarks to the Author):

—
The paper can be accepted in this form

Response: We thank the reviewer for endorsing the publication of our manuscript.